# DockedAC: Empowering Deep Learning Models with 3D Protein-Ligand Data for Activity Cliff Analysis

## Abstract

Artificial intelligence has become a crucial tool in drug discovery, excelling in tasks such as molecular property prediction. An *activity cliff*, which refers to a minor structural modification to a molecule resulting in a large change in its biological activity, poses a challenge in predictive modeling. The activity cliff depends on the interaction between the target and the ligand, which is largely overlooked by previous ligand-centric studies. However, the limited activity cliff data of target-ligand 3D complex restrain the predictive power of modern deep learning models. In this paper, we introduce DockedAC, a new dataset incorporating the protein target and 3D complex structure information for studying the problem of activity cliffs. By matching protein binding information and ligand bioactivity, we employ molecular docking to generate the complex structure for each activity value. The DockedAC dataset contains 82,836 activity data on 52 protein targets with activity cliff annotations, which serves as the first step towards activity cliff research with large-scale 3D complex structures. We benchmark the dataset with traditional machine learning and deep learning approaches. Our data and benchmark platform are available here.

## 1 Introduction

Artificial intelligence (AI) is revolutionizing the drug discovery process as it is capable of large-scale data analysis, pattern recognition, and making accurate predictions (Vamathevan et al., 2019). One important application of AI models is to predict the biological activity of candidate compounds, thereby reducing labor-intensive tasks. A foundational concept in many AI algorithms is the similarity principle, which states that similar objects are likely to share similar features and predictions. However, in drug discovery, a phenomenon called activity cliffs defies this idea and poses a challenge for AI models. An **activity cliff** (AC) is defined as structurally similar compounds exhibiting large differences in their biological activity against the same target (Maggiora, 2006), as illustrated in Figure 1 (a).

AC is crucial for drug discovery, as it complicates the process of optimizing drug candidates by confounding the human experts in the understanding of usual structure-activity relationships (SARs) (Vogt et al., 2011). On the other hand, knowledge about ACs can be highly beneficial when designing or optimizing compounds to enhance the bioactivity of a given target (Cruz-Monteagudo et al., 2014; Stumpfe et al., 2014). For example, replacing a single atom or adding a methyl group can result in more than 100-fold improvement in bioactivity (Leung et al., 2012; Pennington & Moustakas, 2017). However, the mechanisms of ACs in individual drug development programs can be different, making it challenging for humans to process such information and derive transferable experiences. Therefore, various efforts have been made to computationally predict ACs (Stumpfe et al., 2019).

Compared to quantitative structure-activity relationship (QSAR) modeling for other molecular properties, AC predictions are challenging due to the non-robustness that ACs introduce to the models (Cruz-Monteagudo et al., 2016). Early attempts use machine learning methods such as random forest (RF) and support vector machine (SVM) to predict the AC of a compound pair (Guha, 2012; Heikamp et al., 2012). To further improve AC predictions, matched molecular pair (MMP) kernel (Tamura et al., 2021) and condensed graphs of reaction representations (Horvath et al., 2016)

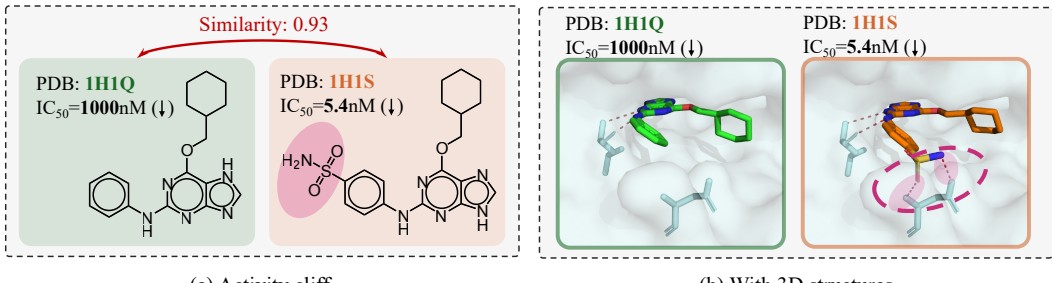

(a) Activity cliff   (b) With 3D structures

Figure 1: Illustration of activity cliffs. (a) An activity cliff example: two similar molecules with a large difference in the bioactivity of the target. (b) From the 3D structure, the bioactivity of the ligand on the right is improved due to the formation of two new hydrogen bonds (highlighted with pink dashed lines).

have been integrated into various machine learning methods. More recently, algorithms based on deep neural networks have been applied to predict ACs, such as convolutional neural networks (Iqbal et al., 2021), graph neural networks (Park et al., 2022) and transformers (Chen et al., 2022).

In most previous works, the study of ACs has been ligand-centric and lacked 3D structure consideration, failing to account for interactions between the ligand and the protein target (Husby et al., 2015; Tamura et al., 2023). Many mechanisms of ACs can be analyzed from the structural perspective, such as hydrogen bonding, ionic interactions, hydrophobic or aromatic group interactions (Hu et al., 2012) (e.g. Figure 1 (b)). It is therefore natural to incorporate the information of structures into the modeling of ACs. However, available structural data for ACs is very limited, with only 215 pairs of AC ligands (Husby et al., 2015). Such data scarcity issue makes it challenging to train deep learning models effectively.

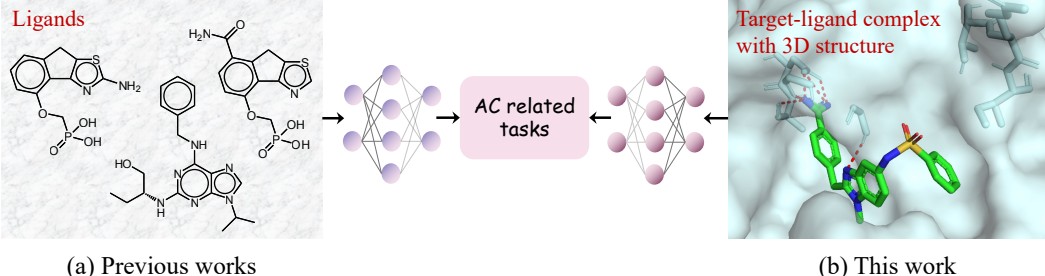

(a) Previous works   (b) This work

Figure 2: Settings of previous studies and our work about ACs. (a) Previous works mostly consider AC prediction from a ligand-centric view and overlook the target information and 3D complex structure. (b) We construct a dataset with target-ligand complex structures for AC prediction.

In this paper, we present DockedAC, a new dataset to tackle the problem of ACs from a structural perspective, aiming at AC modeling with large data and modern AI algorithms. Unlike previous studies, our dataset includes not only the information on protein targets but also the target-ligand complex structures built using molecular docking (Figure 2). We collect the bioactivity data of more than 80,000 ligands across over 50 protein targets. The protein targets are mapped to their corresponding structures in the RCSB Protein Data Bank (PDB) (Berman et al., 2000), with the ligand binding sites identified for docking. In addition, we also provide a framework to benchmark the performance of traditional machine learning and deep learning methods on AC prediction and study the effect of ACs on model performance. Our dataset would be beneficial to enhance model interpretability, inspire the design of promising algorithms on ACs, and foster the development of more effective 3D feature extraction methods.

## 2 RELATED WORK

**Previous works on AC prediction.** As a crucial phenomenon in drug discovery, ACs are not only popular in medicinal chemistry but also attract the attention of the computer science and intelligence community. Various methods of machine learning and deep learning have been applied to the prediction of ACs (Guha, 2012; Heikamp et al., 2012; Iqbal et al., 2021; Chen et al., 2022; Park et al., 2022). In addition, recent research has explored ACs from several different perspectives, such as QSAR modeling (Dablander et al., 2023), the complexity of the learning methods (Tamura et al., 2023), and benchmarking of different approaches (van Tilborg et al., 2022). However, due to the limited availability of data, almost all existing works focus on the ligand-centric view of ACs, where the ligand is modeled with a 2D molecular graph or 1D SMILES sequence (Weininger, 1988), without incorporating the 3D structure and the protein target information. The 3D activity cliff (3DAC) database, used in a study on structure-based AC prediction, contains only 219 3DAC pairs (Hu et al., 2012; Husby et al., 2015). This motivates us to construct a larger dataset for structure-based ACs.

**Existing AC datasets.** Although there are several works on AC prediction, few good benchmarking datasets exist. Several works rely on self-collected datasets and are not well documented, or have little information provided about the protein targets (Jiménez-Luna et al., 2022; Dablander et al., 2023; Tamura et al., 2023). Two recent works on AC datasets both collect data from the ChEMBL database (Mendez et al., 2019), either for the classification of a pair of AC ligands (Zhang et al., 2023c) or the regression of the bioactivity value of individual AC ligands (van Tilborg et al., 2022). These datasets do not consider modeling the 3D structure of the binding complex, rendering them less appropriate for accurate AC prediction. In our work, we match the obtained bioactivity data to the corresponding protein structures in PDB and generate target-ligand binding structures.

**3D protein-ligand binding affinity prediction.** In this work, we consider the regression problem and train different models to predict the bioactivity in the presence of the AC. Given the target-ligand complex structures, nearly all the models for binding affinity prediction use the PDBbind dataset, including convolutional neural networks, graph neural networks, and attention-based models (Zhang et al., 2023a; Jiang et al., 2021a; Jiménez et al., 2018; Tan et al., 2024). A comprehensive review of the drug-target interaction prediction can be found in Zeng et al. (2024). In molecular property prediction, activity cliffs can significantly impact model predictions (Deng et al., 2023). We evaluate the performance of 3D target-ligand affinity prediction models with our dataset and compare them with other machine learning or deep neural network models with ligand-only inputs.

## 3 THE DOCKEDAC DATASET

In summary, the construction of DockedAC involves several key steps: data collection, AC identification, target structure annotation, and target-ligand complex generation. The following section provides a detailed explanation of each step in this process.

### 3.1 DATA COLLECTION

We first collect bioactivity data (Inhibitory Constant, $K_i$; Half-Maximal Effective Concentration, $EC_{50}$; Half-Maximal Inhibitory Concentration, $IC_{50}$ in [nM]) of 64 protein targets from ChEMBL v33 (Mendez et al., 2019) with the ChEMBL web resource client (Davies et al., 2015). To eliminate significant sources of error, the obtained raw data is checked for validity and reliability. In particular, a ligand is removed if (a) it fails the sanitization and standardization by RDkit (Bento et al., 2020); or (b) it has a standard deviation larger than 10 in case of multiple entries. To ensure enough samples of a target for model training, the targets with fewer than 500 ligands are dropped. Finally, the negative logarithm $p$ is applied to the bioactivity values as the regression target (denoted as $pK_i$; $pEC_{50}$; $pIC_{50}$ in [log units]) (Stewart & Watson, 1983). After this process, we have the CHEMBL id of the target and the corresponding ligands with bioactivity values (the first step in Figure 3 (a)). The resulting dataset has 54 protein targets.

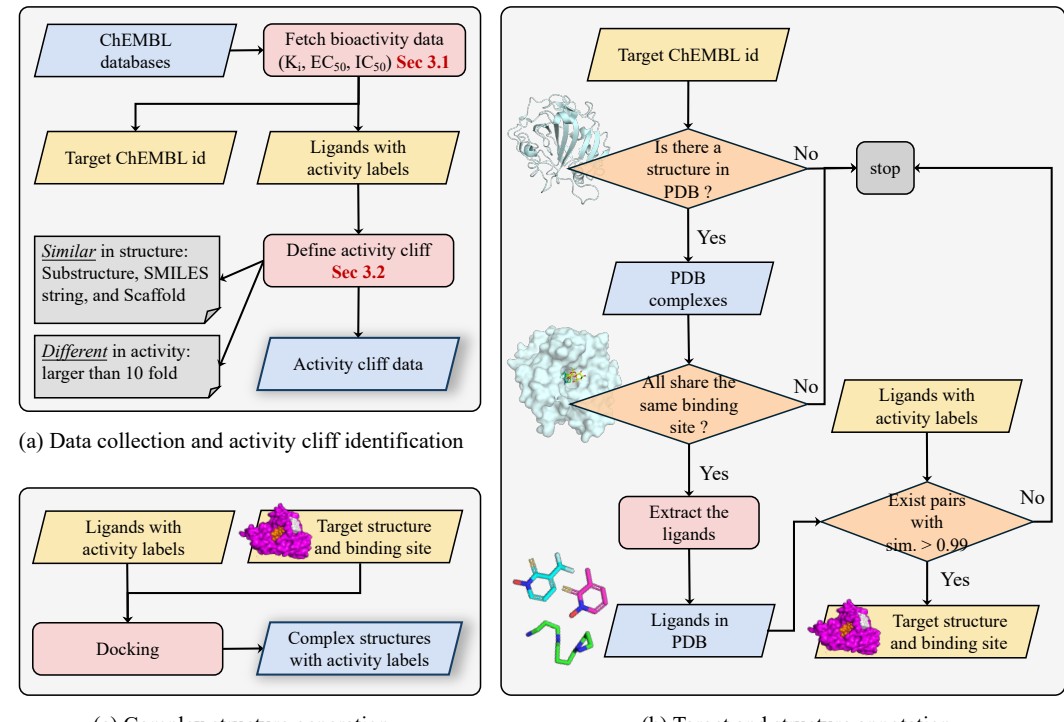

Figure 3: The whole process of building `DockedAC` with: (a) initial data collection from ChEMBL (Sec. 3.1) and activity cliff identification (Sec. 3.2), (b) mapping targets to 3D structures and identifying binding sites (Sec. 3.3), and (c) generation of target-ligand complex structures (Sec. 3.4).

## 3.2 ACTIVITY CLIFF IDENTIFICATION

An activity cliff is a pair of structurally similar compounds with a large difference in bioactivities against a given target. To detect pairs of similar ligands, we take a consensus of three similarity measures to define the activity cliff pairs following van Tilborg et al. (2022): (a) substructure similarity, which is calculated via the Tanimoto coefficient on the extended connectivity fingerprint (ECFP) (Tanimoto, 1958; Rogers & Hahn, 2010); (b) scaffold similarity, which is determined by the Tanimoto coefficient on the ECFP of the generic Murcko scaffolds (Bemis & Murcko, 1996); (c) SMILES similarity, computed as one minus the scaled Levenshtein distance between the canonical SMILES (Levenshtein et al., 1966). If any of these three similarities is equal to or larger than 0.9, the pair of ligands is checked for their difference in bioactivity. Currently, there are no widely accepted quantitative definitions of ACs (Stumpfe et al., 2020). Following previous works (Jiménez-Luna et al., 2022; Hu & Bajorath, 2012), a bioactivity difference larger than one order of magnitude ($10\times$) is used to identify activity cliffs (the second step in Figure 3 (a)).

## 3.3 TARGET AND STRUCTURE ANNOTATION

To generate the target-ligand complex, it is essential to identify the 3D structure of the target protein and its binding site. This mapping process is illustrated in Figure 3 (b). Given a target CHEMBL id, the first step is to map the target protein to its UniProt id (Consortium, 2023) and find all the structures corresponding to the UniProt id in the PDB. We utilize the PDBbind database for initial searching (Wang et al., 2004). If PDBbind does not include the target, we then search for it in the whole PDB. The obtained structures with a small molecule ligand are chosen and aligned to check if the ligands bind to the same site. If the binding site is not unique, the target is discarded (see Figure 9 (a)(b)). After alignment, ligands sharing the same binding site are extracted and compared with the ligands that have activity labels from ChEMBL. Suppose there exists a pair of ligands, one from the PDB database and one from the ChEMBL database, with a similarity (Tanimoto coefficient of the fingerprints) larger than 0.99. In that case, the target structure and the binding site are used.

Otherwise, the target is removed from the dataset. When multiple structures satisfy this condition, the structure with the best resolution is selected. This procedure ensures the correspondence between the bioactivity values and the target binding site. As a result of this structure mapping process, two targets are removed, resulting in a final dataset of 52 protein targets.

### 3.4 COMPLEX STRUCTURE GENERATION

Next, molecular docking is employed to generate the target-ligand complex for each target, illustrated in Figure 3 (c). The docking tool DSDP is used, which combines the pose sampling algorithm of AutoDock Vina and GPU acceleration (Huang et al., 2023; Trott & Olson, 2010). Since the binding site information of the target is already known, local docking is performed within the given binding region of a 25 Å wide box. The docking results are reviewed to ensure the agreement between the ligand bioactivity value and the binding conformation. A docking score (in kcal/mol) larger than zero indicates an inaccurate docking conformation (e.g. Figure 9 (c)), and the corresponding ligand is removed from the dataset.

### 3.5 DATASET SPLITTING

To prepare the dataset for benchmarking, the ligands of each target are split into a training and test set using a double-stratified sampling strategy (van Tilborg et al., 2022). In particular, the ligands of each target are first clustered into 5 groups based on their substructural similarity (Tanimoto similarity of the ECFP). A two-stage stratified splitting (80%/20%) is then performed on the cluster label and the AC label. This procedure ensures that the training and test set have similar ligand distributions.

### 3.6 DATASET DESCRIPTION

The final dataset contains 82,836 target-ligand activity values and the corresponding generated complex structures. Due to the page limit, the detailed information on each target can be found in Appendix Table 3. We give a brief dataset description in Table 1. The dataset contains popular target families in drug discovery (G-protein-coupled receptors (GPCR), kinases, proteases, and nuclear receptors) as well as targets with critical roles in biology (chaperone and kinesin). In terms of size, the target Carbonic anhydrase II has the

Table 1: Brief dataset statistics by the target type.

| Target type | # Targets | Avg. # ligands | %AC |
|---|---|---|---|
| G protein-coupled receptor | 12 | 2091 | 41.7 |
| Kinase | 11 | 1234 | 27.5 |
| Protease | 8 | 1667 | 38.0 |
| Nuclear receptor | 8 | 1299 | 35.7 |
| Phosphodiesterase | 3 | 1328 | 34.1 |
| Phosphatase | 2 | 1581 | 18.0 |
| Transporter | 1 | 1051 | 25.3 |
| Transferase | 1 | 960 | 41.8 |
| Oxidoreductase | 1 | 739 | 38.0 |
| Other membrane receptor | 1 | 1328 | 38.2 |
| Lyase | 1 | 5796 | 42.2 |
| Kinesin | 1 | 719 | 43.2 |
| Electrochemical transporter | 1 | 1702 | 37.5 |
| Chaperones | 1 | 999 | 15.7 |

most ligands with bioactivity values (5794 unique molecules). The target with the least ligands (533 unique molecules) is Matrix metalloproteinase 8. As an intensively studied drug target, the GPCR is the target family with the most ligands on average. For all the targets, around 37% of the ligands are annotated as ACs, with percentages ranging from 15.7% to 43.2%.

## 4 BENCHMARK

In addition to the `DockedAC` dataset, we also provide a framework to benchmark the performance of various machine learning and deep learning methods on AC prediction. This section briefly introduces our benchmark setup, followed by a detailed presentation of the experimental results and analyses in the subsequent section (Sec. 5).

### 4.1 MODEL DESCRIPTIONS

In general, three types of learning models are included in our framework:

- Four classic machine learning algorithms for structure-activity relationship prediction using handcrafted molecular descriptors: K-nearest neighbor (KNN) (Cover & Hart, 1967), random forest (RF) (Breiman, 1996), gradient boosting machine (GBM) (Friedman, 2001), and support vector regression (SVM) (Hearst et al., 1998).
- Deep learning models that only leverage the 1D or 2D ligand information, including (1) three 1D sequential models: transformer (Vaswani et al., 2017), long short-term memory (LSTM) networks (Hochreiter & Schmidhuber, 1997), and 1D CNN (Kimber et al., 2021), and (2) four 2D structural graph neural network (GNN) models: message passing neural network (MPNN) (Gilmer et al., 2017), graph convolutional network (GCN) (Kipf & Welling, 2016), graph attention network (GAT) (Vaswani et al., 2017), and attentive fingerprint (AFP) (Xiong et al., 2019).
- Two 3D structural GNN models: IGN (Jiang et al., 2021a) and SS-GNN (Zhang et al., 2023a) are included to study the effect of 3D structures, as our dataset contains 3D structural information.

## 4.2 Feature Descriptions

For machine learning algorithms, following previous work van Tilborg et al. (2022), we consider four types of molecule descriptors from several levels of complexity as follows. (1) Extended Connectivity Fingerprints (ECFPs) (Rogers & Hahn, 2010): circular topological fingerprints used for molecular characterization. (2) Molecular ACCess System (MACCS) keys (Durant et al., 2002): a set of structural keys utilized for substructure searching and similarity analysis, encoding specific chemical substructures or patterns. (3) Physicochemical (PhysChem) descriptors (Walters & Murcko, 2002): 11 properties indicative of drug-likeness. (4) Weighted Holistic Invariant Molecular (WHIM) descriptors (Todeschini et al., 1998): capturing three-dimensional geometrical and electronic properties of molecules, invariant to rotation and translation.

Deep learning methods eliminate the need for handcrafted descriptors, allowing direct learning from "unstructured" data representations. For sequential methods, the Simplified Molecular Input Line Entry System (SMILES) (Weininger, 1988) string is used, which is popular for its ability to describe the structure of chemical species in text format that sequential methods can naturally process. For 2D GNN models, we adopt molecular graphs, a representation of structural formula where nodes represent atoms and edges represent bonds. For 3D GNN models, we employ the target-ligands complexes we have processed that incorporate detailed 3D structure information. Detailed descriptions of the features can be found in Appendix A.4.

## 4.3 Metrics and Implementations

For each target, we train separate regression models on the bioactivity values ($pK_i/pEC_{50}/pIC_{50}$ in [log units]). The regression setting makes it possible to compare the AC and non-AC tasks. The root-mean-square error (RMSE) is employed as the evaluation metric to quantify the performance. The RMSE represents the error calculated across all ligands, whereas $RMSE_{cliff}$ specifically denotes the error computed for AC ligands. For model implementation, we conduct hyperparameter tuning through grid search and report the results from five-fold cross-validation. Further details on these methods and their implementations are provided in Appendix A.3 and A.5.

## 5 Experimental Results and Analyses

### 5.1 Performance Comparison for GNN Models

To investigate the effect of 3D structure information, we first evaluate 2D GNN models and 3D GNN models across 52 targets. To study AC, scatter plots with RMSE as the x-axis and $RMSE_{cliff}$ as the y-axis are utilized, as shown in Figure 4 (a) to (f).

We have the following empirical observations: 1) The majority of the points are distributed above the line RMSE = $RMSE_{cliff}$, indicating higher prediction errors on ACs due to their unusual structure-activity relationships. 2) Despite a general correlation between RMSE and $RMSE_{cliff}$, notable outliers are presented. This suggests that models with overall high prediction accuracy do not necessarily perform well on ACs. Among these models, SS-GNN exhibits the closest distribution around line RMSE = $RMSE_{cliff}$, with only two targets deviating by more than 0.2 log units. 3) The distribution of IGN is primarily clustered in the lower-left corner of the plots, indicating superior performance in

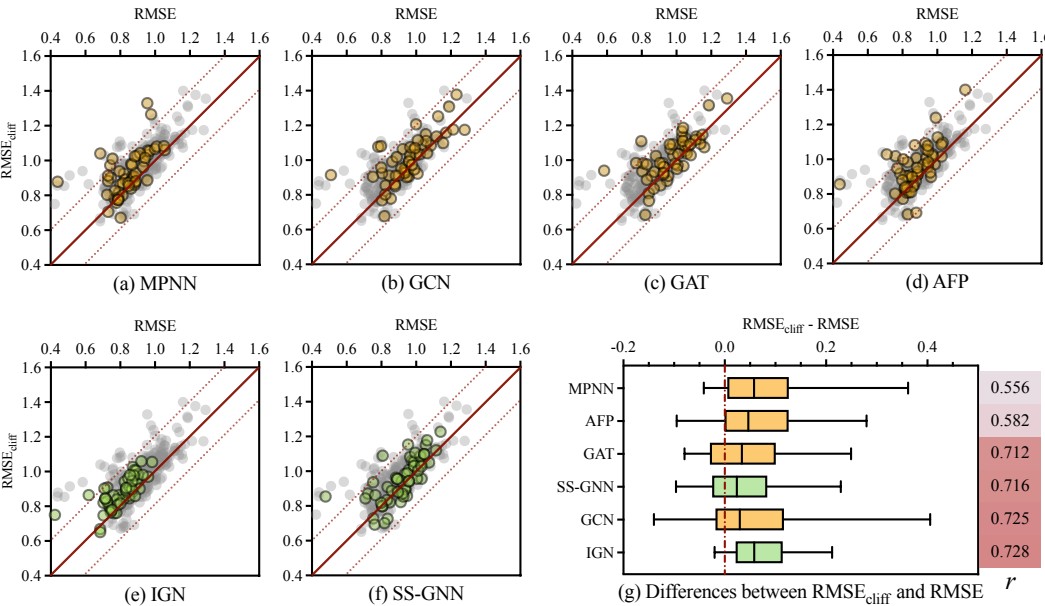

Figure 4: Performance comparison for GNN models. (a)-(f) Comparison between RMSE and $\text{RMSE}_{\text{cliff}}$ of GNN models across 52 targets. The 2D GNN models are colored in  yellow , while the 3D GNN models are colored in  green . Gray nodes depict all nodes in these six subgraphs for a clear comparison. Red solid lines show RMSE = $\text{RMSE}_{\text{cliff}}$, while red dashed lines indicate a $\pm 0.2$ log units difference. (g) Target-wise differences between overall RMSE and $\text{RMSE}_{\text{cliff}}$ for all GNN models ordered by Pearson correlation $r$ of RMSE and $\text{RMSE}_{\text{cliff}}$.

both RMSE and $\text{RMSE}_{\text{cliff}}$. This suggests that incorporating 3D structural information enhances the prediction of ACs and improves the model's understanding of standard structure-activity relationships. 4) Figure 4 (g) further presents the target-wise differences between RMSE and $\text{RMSE}_{\text{cliff}}$ for GNN models, sorted by the Pearson correlation coefficient $r$ of RMSE and $\text{RMSE}_{\text{cliff}}$. 3D structure GNN models ranked first and third in terms of $r$. SS-GNN exhibits the smallest difference between RMSE and $\text{RMSE}_{\text{cliff}}$, while IGN has the most concentrated distribution across targets. Its 5%-95% coverage range is only 0.58 times that of MPNN and 0.71 times that of GAT. These findings demonstrate the benefit of incorporating 3D structural information, which leads to a higher degree of correlation between performance on overall ligands and AC ligands, ultimately improving the understanding of structure-activity relationships and aiding in the prediction of ACs.

Table 2: The $\text{RMSE}_{\text{cliff}}$ evaluated using GNN models and machine learning algorithms with ECFP featurization across the top four target families. For each method, the colors show the ranking of the target, i.e.,  first ,  second ,  third ,  fourth .

| Target type (#) | MPNN | GCN | GAT | AFP | IGN | SS-GNN | KNN | RF | GBM | SVM |
|---|---|---|---|---|---|---|---|---|---|---|
| GPCR (12) | 0.927 | 0.995 | 1.018 | 0.907 | 0.877 | 0.977 | 0.814 | 0.785 | 0.791 | 0.752 |
| Kinase (11) | 0.902 | 0.942 | 0.970 | 0.917 | 0.865 | 0.896 | 0.802 | 0.765 | 0.747 | 0.707 |
| Protease (8) | 0.979 | 1.071 | 1.069 | 1.025 | 0.904 | 1.006 | 0.867 | 0.827 | 0.828 | 0.810 |
| Nuclear receptor (8) | 0.893 | 0.972 | 0.978 | 0.932 | 0.865 | 0.906 | 0.822 | 0.799 | 0.800 | 0.781 |

## 5.2 THE AC PREDICTION IS TARGET-DEPENDENT

The AC effect is determined by the interaction between the ligand and the target. We hypothesize that the target type may also influence deep learning model performance. Table 2 shows the average $\text{RMSE}_{\text{cliff}}$ of the top four target families that have the most targets in our dataset, i.e., GPCR, kinase, protease, and nuclear receptor. The color means the ranking of the four targets for each method. It is

easy to notice that performance rankings are quite consistent across both deep learning and machine learning methods. Protease has the worst $RMSE_{cliff}$ for all the methods while kinase is the target family with the best $RMSE_{cliff}$ for most methods. GPCR has a worse performance than nuclear receptor in most deep learning methods, but machine learning methods perform better on GPCR.

It is also surprising to observe better performance for some machine learning models over deep learning approaches. This can be attributed primarily to their use of handcrafted features, especially ECFP. To validate this observation, we implement a hybrid approach combining the ECFP features with the features extracted from the last layer of the 3D IGN model. These concatenated features are then fed into an MLP for prediction (as illustrated in Appendix Figure 8). The promising results across ten targets (see Appendix Table 6) demonstrate the effectiveness of ECFP in structure-activity relationship learning. On the other hand, this experiment underscores the value of integrating traditional cheminformatics techniques with advanced deep learning methods in molecular property prediction tasks. Future research could explore optimizing this hybrid approach and investigating its applicability to a broader range of molecular targets and properties.

### 5.3 THE PERCENTAGE OF AC MATTERS

In general, machine learning models tend to perform better with more training data. Here we study the factors influencing the AC prediction. Surprisingly, we do not find that the number of training samples produces a significant correlation with RMSE, $RMSE_{cliff}$, or their numerical difference, i.e., $RMSE_{cliff} - RMSE$ (see Appendix Figure 11). However, as shown in Figure 5 (also in Figure 10), the ratio of AC ligands in the training set is a significant factor affecting $RMSE_{cliff} - RMSE$, with a p-value of 1.0e-4. A higher percentage of the AC in the training set means more information about AC, thus improving the AC predictive power. Our finding indicates that the knowledge about general bioactivity prediction is different from the knowledge benefiting AC

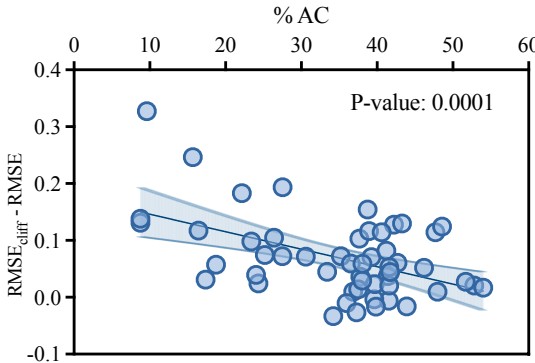

Figure 5: Relationship between the ratio of the AC and $RMSE_{cliff} - RMSE$ of IGN.

prediction, underlining the importance of new datasets and methods tailored for AC prediction.

### 5.4 PERFORMANCE COMPARISON WITH MACHINE LEARNING ALGORITHMS

We benchmark the ability of all methods to predict bioactivity in the presence of the AC (measured by $RMSE_{cliff}$), as shown in Figure 6 (detailed results in Appendix Figure 12). We have the following empirical observations: 1) Significant performance differences can be observed among targets in the handling of AC compounds, with $RMSE_{cliff}$ values spanning from 0.52 to 1.59 log units, which is consistent with previous works (van Tilborg et al., 2022; Sheridan, 2012). This highlights the challenges of AC prediction and the necessity for further development of advanced algorithms and more effective feature extraction methods. 2) Among the four machine learning algorithms, performance disparities primarily stem from the molecule descriptor rather than the learning methods. Nonbinary descriptors such as WHIM and PhysChem significantly underperform compared to ECFP. ECFPs are designed specifically for structure-activity modeling by encoding detailed information about each atom's local environment, yielding the lowest prediction error of all methods. Its strong discriminative capability effectively differentiates molecules, even with minor structural differences. This effectiveness is further corroborated by the promising results obtained when combining ECFP with 3D graph models (as detailed in Appendix Figure 8 and Table 6). 3) For deep learning methods, IGN coupled with 3D structure information achieves the best performance on ACs. This approach benefits from the interaction information between the ligand and the protein target captured within the 3D structure.

### 5.5 PERFORMANCE POSITIONING OF 3D GNN METHODS

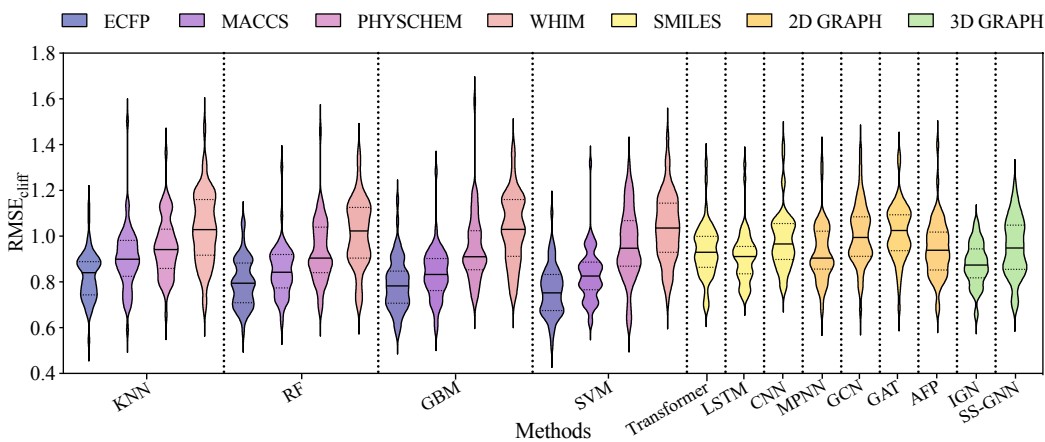

Figure 6: The RMSE$_{cliff}$ evaluated using different methods and features across 52 targets.

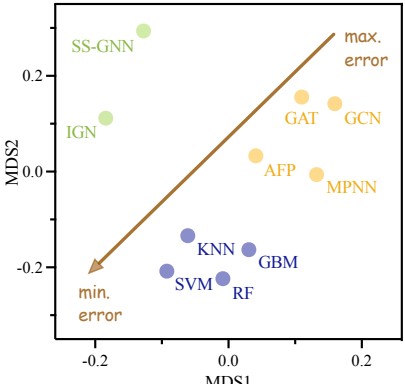

Our experimental results show that machine learning methods significantly outperform deep learning methods, especially with the ECFP featurization. This finding aligns with previous studies on molecular property prediction (Jiang et al., 2021b; Janela & Bajorath, 2022). To provide a global assessment of the methods and demonstrate the effect of the target on 3D GNN methods, we take the RSME$_{cliff}$ values of the 52 targets as features and compute the Pearson correlation between the methods. The correlation serves as a similarity measure in multidimensional scaling (MDS) (Mead, 1992) to visualize the methods in a 2D plane (Figure 7). We then identify the direction that determines the performance of the methods. Although the average performance of SS-GNN and IGN does not surpass machine learning methods, there exist specific targets that IGN and SS-GNN outperform SVM. In contrast, GAT and GCN consistently have larger RSME$_{cliff}$ values than SVM across all targets. Handcraft features, such as ECFP, have been

Figure 7: MDS visualization of GNN-based models and machine learning algorithms with ECFP featurization on RMSE$_{cliff}$.

optimized for QSAR over decades. Our analysis indicates that the models with 3D structures can offer insights that handcraft features do not capture. Therefore, in practice, models based on 3D structures can be important complements to the machine learning methods.

The incorporation of 3D structural information also enables cross-target modeling capabilities. To investigate this potential, we conduct additional experiments by combining $K_i$ targets and training the IGN model under two scenarios: (i) in-domain setting under all $K_i$ targets, and (ii) out-of-distribution (OOD) setting excluding Protease-type targets. Analysis of four Protease targets (Appendix Table 7) reveals that the absence of Protease targets in training leads to performance degradation, with average RMSE$_{cliff}$ increasing from 0.9 to 1.4. While multi-target training achieves comparable performance to target-specific training across all targets (Appendix Figure 13), these results indicate that there is still a long way to go to fully exploit the multi-target 3D data and make the model generalize to new targets.

## 6 CONCLUSION

In this paper, we introduce DockedAC, a new dataset for ACs with 3D complex structures. The dataset contains over 80k ligands from 52 protein targets, with the 3D structure of each target annotated by a unique known binding site. Molecular docking is performed to generate the protein-ligand complex structures for at least 500 ligands per target. We benchmark the dataset with various

machine learning and deep learning methods, finding that for GNN-based methods, introducing 3D information can enhance AC prediction and reduce the gap between general and AC activity prediction. Our experiments suggest that the absolute error of AC prediction is target-dependent, and the ratio of AC ligands in the training set is an important factor influencing the difference between general and AC activity prediction. In addition, deep learning methods cannot compete with traditional machine learning methods using fingerprints, highlighting the need to develop new 3D QSAR algorithms. `DockedAC` serves as a first step in this direction from the perspective of 3D complex structures and target-ligand interactions.

**Limitations.** While our dataset contains a variety of protein targets, the distribution of different types of targets is imbalanced, with several popular drug targets dominating. Diversifying the target types is beneficial to improve the generalization of successive models. Furthermore, the mapping between the target and the unique binding site may introduce bias, as some targets have unknown binding sites. We plan to conduct routine validity checks to update the dataset as more protein structures are deposited into the PDB. Lastly, the complex structures generated by molecular docking may be inaccurate, and more advanced approaches such as molecular simulation can be employed to refine the complex structures.

**Future Work.** `DockedAC` provides the foundation for studying ACs from a structural perspective, and we anticipate that it will inspire the development of novel 3D QSAR algorithms. Future research could focus on designing deep learning architectures that effectively capture and utilize 3D structural information to improve AC prediction accuracy. Additionally, the dataset could be expanded to include more diverse targets and ligands, as well as refined complex structures, to further enhance its value for AI-driven drug discovery. We believe that `DockedAC` dataset will foster the development of innovative computational methods and contribute to the advancement of rational drug design.

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

# A APPENDIX

# A DATASETS AND BASELINE MODELS

## A.1 LICENSE AND AVAILABILITY

The code for benchmark is available here: https://anonymous.4open.science/r/DockedAC-ICLR/README.md. The DockedAC dataset and its future updates can be found here: https://doi.org/10.5281/zenodo.11485280.

The DockedAC dataet is licensed under Creative Commons Attribution-ShareAlike 4.0 International License. For details, please see https://creativecommons.org/licenses/by-sa/4.0/. The content of DockedAC uses data from RCSB PDB and ChEMBL. The PDB archive are available under the CC0 1.0 Universal (CC0 1.0) Public Domain Dedication (https://creativecommons.org/publicdomain/zero/1.0/). ChEMBL is provided under a Creative Commons Attribution-ShareAlike 3.0 Unported license (https://creativecommons.org/licenses/by-sa/3.0/).

## A.2 DATASETS

Our introduced dataset DockedAC[1] comprises 82,836 ligands from 52 protein targets, which is meticulously curated to support various machine learning and deep learning studies related to activity cliff (AC) prediction. Table 3 provides detailed statistics of DockedAC.

## A.3 BASELINE MODELS

In this work, we integrate 13 recent baselines commonly used for structure-activity relationship prediction, including four traditional machine learning algorithms: KNN, RF, GBM, and SVM; three sequential models: LSTM, Transformer, and 1D CNN; four 2D GNN models: GCN, GAT, MPNN, and AFP; and two 3D structure GNN models: IGN and SS-GNN. The detailed descriptions of these approaches are listed in the following:

- **KNN** (Cover & Hart, 1967). K-Nearest Neighbor (KNN) is a simple, non-parametric method that predicts the target molecule's response by averaging the response of the k-nearest neighbors from the training set.
- **RF** (Breiman, 1996). Random Forest (RF) is an ensemble method that combines the outputs of multiple decision trees to improve accuracy and reduce over-fitting. Each decision tree is built upon a subset of the training set, and the final prediction is obtained by averaging the results from these individual trees.
- **GBM** (Friedman, 2001). Similar to RF, Gradient Boosting Machine (GBM) also combines the predictions of multiple decision trees. However, in GBM, these trees are built sequentially, with each subsequent tree specially designed to correct the errors of its predecessors.
- **SVM** (Hearst et al., 1998). Support Vector Machine (SVM) aims to identify a linear regression plane in a higher-dimensional space created by applying a designated kernel function. In this work, the Radial Basis Function (RBF) kernel is used.
- **Transformer** (Vaswani et al., 2017). The Transformer model leverages self-attention mechanisms to capture dependencies across different positions in the input sequence. In our work, we employed the pretrained ChemBERTa (Chithrananda et al., 2020) architecture, which has been trained on 10 million compounds.
- **LSTM** (Hochreiter & Schmidhuber, 1997). Long Short-Term Memory (LSTM) can capture temporal dependencies and patterns in sequential data by maintaining long-term memory through their gated structure. In this work, we employ SMILES strings as the input for the model.
- **1D CNN** (Kimber et al., 2021). Convolutional Neural Network (CNN) uses convolutional filters to aggregate spatial information from adjacent positions. For processing sequential SMILES string data, we employ 1D CNNs that perform convolutional operations along a single dimension.
- **MPNN** (Gilmer et al., 2017). Message Passing Neural Network (MPNN) operates by iteratively passing messages between nodes and updating their representations based on neighboring nodes.

---

[1]https://anonymous.4open.science/r/DockedAC-ICLR

Table 3: Dataset overview. n (where $n_{\text{train}}/n_{\text{test}}$, *resp.*) represents the total number of compounds, divided into training and test sets. $n^{AC}$ (where $n_{\text{train}}^{AC}$ / $n_{\text{test}}^{AC}$ *resp.*) denotes the total number of activity cliff compounds within the dataset, also divided into training and test sets.

| Target Name | ChEMBL ID | PDB | Type | n ($n_{\text{train}}$ / $n_{\text{test}}$) | $n^{AC}$ ($n_{\text{train}}^{AC}$ / $n_{\text{test}}^{AC}$) |
|---|---|---|---|---|---|
| Androgen Receptor | CHEMBL1871 | 2ama | $K_i$ | 617 (492/125) | 135 (109/26) |
| Cannabinoid CB1 receptor | CHEMBL218 | 6kqi | $EC_{50}$ | 1004 (802/202) | 369 (293/76) |
| Coagulation factor X | CHEMBL244 | 2p93 | $K_i$ | 3093 (2474/619) | 1476 (1180/296) |
| Delta opioid receptor | CHEMBL236 | 6pt3 | $K_i$ | 2580 (2060/520) | 1005 (802/203) |
| Dopamine D3 receptor | CHEMBL234 | 3pbl_A | $K_i$ | 3657 (2924/733) | 1604 (1284/320) |
| Dopamine D4 receptor | CHEMBL219 | 5wiu_A | $K_i$ | 1865 (1491/374) | 740 (592/148) |
| Dopamine transporter | CHEMBL238 | 2q6h_A | $K_i$ | 1051 (838/213) | 266 (211/55) |
| Dual specificity protein kinase CLK4 | CHEMBL4203 | 6fyv | $K_i$ | 731 (582/149) | 64 (51/13) |
| Bile acid receptor FXR | CHEMBL2047 | 5q0u | $EC_{50}$ | 631 (503/128) | 245 (195/50) |
| Ghrelin receptor | CHEMBL4616 | 6ko5_A | $EC_{50}$ | 673 (534/139) | 355 (282/73) |
| Glucocorticoid receptor | CHEMBL2034 | 4lsj | $K_i$ | 684 (551/133) | 243 (194/49) |
| Glycogen synthase kinase-3 beta | CHEMBL262 | 6hk3 | $K_i$ | 855 (683/172) | 160 (128/32) |
| Histamine H1 receptor | CHEMBL231 | 3rze_A | $K_i$ | 972 (776/196) | 237 (189/48) |
| Histamine H3 receptor | CHEMBL264 | 7f61_A | $K_i$ | 2862 (2288/574) | 1191 (952/239) |
| Tyrosine-protein kinase JAK1 | CHEMBL2835 | 4k77 | $K_i$ | 615 (489/126) | 60 (47/13) |
| Tyrosine-protein kinase JAK2 | CHEMBL2971 | 4jia | $K_i$ | 976 (779/197) | 162 (128/34) |
| Kappa opioid receptor | CHEMBL237 | 4djh | $EC_{50}$ | 953 (761/192) | 456 (365/91) |
| Kappa opioid receptor | CHEMBL237 | 4djh | $K_i$ | 2599 (2078/521) | 1109 (887/222) |
| Orexin receptor 2 | CHEMBL4792 | 5wqc | $K_i$ | 1471 (1174/297) | 794 (634/160) |
| Peroxisome proliferator-activated receptor alpha | CHEMBL239 | 3kdu | $EC_{50}$ | 1721 (1374/347) | 699 (558/141) |
| Peroxisome proliferator-activated receptor delta | CHEMBL3979 | 5xmx | $EC_{50}$ | 1125 (899/226) | 468 (374/94) |
| Peroxisome proliferator-activated receptor gamma | CHEMBL235 | 2yfe | $EC_{50}$ | 2349 (1877/472) | 885 (707/178) |
| PI3-kinase p110-alpha subunit | CHEMBL4005 | 6gvf | $K_i$ | 960 (767/193) | 401 (320/81) |
| Serine/threonine-protein kinase PIM1 | CHEMBL2147 | 2j2i | $K_i$ | 1456 (1162/294) | 572 (456/116) |
| Serotonin 1a (5-HT1a) receptor | CHEMBL214 | 7e2x_R | $K_i$ | 3317 (2651/666) | 1222 (977/245) |
| Serotonin transporter | CHEMBL228 | 6awo_A | $K_i$ | 1702 (1362/340) | 638 (511/127) |
| Sigma opioid receptor | CHEMBL287 | 6dk1 | $K_i$ | 1328 (1061/267) | 507 (404/103) |
| Thrombin | CHEMBL204 | 1mu8 | $K_i$ | 2747 (2195/552) | 1089 (870/219) |
| Tyrosine-protein kinase ABL | CHEMBL1862 | 2hzi | $K_i$ | 794 (633/161) | 330 (263/67) |
| Mu opioid receptor | CHEMBL233 | 8feo_R | $K_i$ | 3141 (2511/630) | 1294 (1035/259) |
| Cyclin-dependent kinase 2 | CHEMBL301 | 1h1q | $IC_{50}$ | 1454 (1161/293) | 350 (279/71) |
| Serine/threonine-protein kinase Chk1 | CHEMBL4630 | 2brb | $IC_{50}$ | 1701 (1359/342) | 826 (660/166) |
| 3-phosphoinositide dependent protein kinase-1 | CHEMBL2534 | 1uu3 | $IC_{50}$ | 705 (562/143) | 282 (224/58) |
| Phosphodiesterase 5A | CHEMBL1827 | 4ia0 | $IC_{50}$ | 1609 (1285/324) | 667 (532/135) |
| Dihydrofolate reductase | CHEMBL202 | 1u71 | $IC_{50}$ | 739 (590/149) | 281 (223/58) |
| Urokinase-type plasminogen activator | CHEMBL3286 | 1owe | $K_i$ | 718 (572/146) | 191 (151/40) |
| Carbonic anhydrase II | CHEMBL205 | 5sz6 | $K_i$ | 5796 (4636/1160) | 2444 (1957/487) |
| Estrogen receptor alpha | CHEMBL206 | 1qkt | $IC_{50}$ | 2094 (1674/420) | 700 (559/141) |
| Heat shock protein HSP 90-alpha | CHEMBL3880 | 4o0b | $IC_{50}$ | 999 (797/202) | 157 (125/32) |
| Fructose-1,6-bisphosphatase | CHEMBL3975 | 2jjk | $IC_{50}$ | 556 (443/113) | 153 (122/31) |
| Protein-tyrosine phosphatase 1B | CHEMBL335 | 1nny | $IC_{50}$ | 2607 (2084/523) | 229 (183/46) |
| Matrix metalloproteinase 8 | CHEMBL4588 | 3dng | $IC_{50}$ | 533 (425/108) | 163 (130/33) |
| Dipeptidyl peptidase IV | CHEMBL284 | 2ole | $IC_{50}$ | 2507 (2003/504) | 691 (551/140) |
| Vascular endothelial growth factor receptor 2 | CHEMBL279 | 3vhk | $K_i$ | 780 (622/158) | 135 (108/27) |
| Matrix metalloproteinase 13 | CHEMBL280 | 4jpa | $IC_{50}$ | 2112 (1688/424) | 976 (780/196) |
| Methionine aminopeptidase 2 | CHEMBL3922 | 6qef | $IC_{50}$ | 565 (450/115) | 193 (154/39) |
| Kinesin-like protein 1 | CHEMBL4581 | 5zo8 | $IC_{50}$ | 719 (573/146) | 311 (248/63) |
| Beta-secretase 1 | CHEMBL4822 | 4h3j | $K_i$ | 1061 (847/214) | 549 (438/111) |
| Phosphodiesterase 4B | CHEMBL275 | 3w5e | $IC_{50}$ | 1432 (1143/289) | 535 (426/109) |
| Phosphodiesterase 4D | CHEMBL288 | 2qyn | $IC_{50}$ | 942 (752/190) | 220 (176/44) |
| MAP kinase p38 alpha | CHEMBL260 | 2zbl | $IC_{50}$ | 3502 (2799/703) | 1333 (1065/268) |
| Estrogen receptor beta | CHEMBL242 | lzaf | $IC_{50}$ | 1176 (937/239) | 425 (337/88) |

- **GCN** (Kipf & Welling, 2016). Graph Convolutional Network (GCN) performs convolution operations on graphs.
- **GAT** (Vaswani et al., 2017). Graph Attention Network (GAT) introduces attention mechanisms to GNN to weigh the importance of different neighbors.
- **AFP** (Xiong et al., 2019). Attentive Fingerprint (AFP) employs attention mechanisms at both the atom and molecule levels to learn local and nonlocal properties, enabling it to capture substructural details effectively.
- **IGN** (Jiang et al., 2021a). IGN models the molecular interactions in 3D space. In IGN, two graph convolution modules are layered to learn intramolecular interactions and then sequentially intermolecular interactions.
- **SS-GNN** (Zhang et al., 2023a). Like IGN, SS-GNN is also a 3D structure GNN model tailored for affinity prediction. It constructs a 3D structure graph for protein-ligand interactions based on a

distance threshold, reducing both the graph data scale and computational cost by omitting covalent bonds in proteins.

### A.4  MODEL FEATURES.

In addition to the molecular descriptor used for machine learning algorithms (introduced in Sec. 4.2), we further delve into the featurization for deep learning models. Detailed information on all featurizations and the corresponding models used can be found in Table 4.

For sequential methods, SMILES strings were encoded as one-hot vectors, with truncation applied to strings exceeding 200 characters. To enhance model robustness, tenfold data augmentation was applied using up to nine additional noncanonical SMILES strings for each SMILES string in the dataset, generated via RDKit (Landrum et al., 2013).

For 2D GNN methods, the node has the following features: atom type (one-hot), atomic vertex degree (one-hot), orbital hybridization (one-hot), aromaticity (one-hot), atomic weight (float), formal charge (integer), number of radical electrons (integer), and number of connected hydrogens (integer). For MPNN and AFP, two one-hot bond features are used for the edges, i.e., the bond type and conjugation.

For SS-GNN, there are 11 node features, including atom type, formal charge, hybridization, atom valence, atom degree, number of hydrogens, chirality, atomic mass, aromatic, atom coordinates, and whether belonging to the protein. The edge features include covalent bond type, aromatic, bond length, bond direction, bond stereochemistry, and edge type. The atom coordinates and bond length are extracted from the 3D structures. Further details can be found in Zhang et al. (2023a).

For IGN, it uses similar 2D node and edge features. In addition, IGN uses four new edge features from the 3D structures, including bond length, angle statistics, area statistics, and distance statistics. For detailed descriptions of the features, see Jiang et al. (2021a).

Table 4: Featurization and corresponding baseline models.

| Featurization | Baseline Models | Augmentation |
|---|---|---|
| ECFP Descriptor | KNN, RF, GBM, SVM, | ✗ |
| MACCS Descriptor | KNN, RF, GBM, SVM, | ✗ |
| PHYSCHEM Descriptor | KNN, RF, GBM, SVM, | ✗ |
| WHIM Descriptor | KNN, RF, GBM, SVM, | ✗ |
| SMILES string | LSTM, Transformer, 1D CNN | ✔ 10 times |
| 2D GRAPH | MPNN, GCN, GAT, AFP | ✗ |
| 3D GRAPH | IGN, SS-GNN | ✗ |

### A.5  ADDITIONAL EXPERIMENTAL DETAILS

**Hardware Specifications.** All our experiments were carried out on an NVIDIA RTX3090 GPU with 24G memory. The training time of a target for MPNN, GAT, GCN, and AFP is around 0.5 hours. Training of one target takes around 1 hour and 4 hours for SS-GNN and IGN, respectively.

**Implementation Details.** Traditional machine learning algorithms including KNN, SVM, GBM, and RF regression models were implemented using the Scikit-Learn library[2].

Deep learning algorithms were trained for 500 epochs with early stopping, set with patience of 10 epochs. Four GNN models are implemented using the PyTorch Geometric package[3]. For the MPNN, GCN, and GAT, global pooling was enabled using a graph multiset Transformer (Baek et al., 2021) with eight attention heads, followed by a fully connected prediction head. Each of these models utilized two graph layers. The Transformer model was based on the ChemBERTa (Chithrananda et al., 2020) architecture, using weights derived from 10M compounds in PubChem. Fine-tuning was conducted by freezing the original model weights and substituting the final pooling layer with a regression head. Following van Tilborg et al. (2022), the LSTM model is pretrained on the SMILES

---

[2]https://scikit-learn.org/
[3]https://www.pyg.org/

strings with the next token prediction objective. For the SS-GNN model, we conducted a pretraining phase on the original dataset, PDBbind V2019 (Wang et al., 2004; 2005). In contrast, the IGN model was not fine-tuned using the original dataset due to a mismatch in the model dimensions caused by the varying types of atoms in the dataset. Consequently, we opted to train the IGN model from scratch.

**Hyperparameter Optimization.** Hyperparameter optimization was conducted through grid search. Hyperparameter combinations were evaluated for all models using five-fold cross-validation. Table 5 shows the detailed hyperparameter search space.

Table 5: Hyperparameter search space.

| Methods | Hyperparameters | Search Space |
|---|---|---|
| KNN | The number of nearest neighbors, $k$ | $k = [3, 5, 11, 21]$ |
| RF | The number of trees, $n_t$ | $n_t = [100, 250, 500, 1000]$ |
| GBM | The number of boosting stages, $n_b$
The maximum depth of the model, $n_d$ | $n_b = [100, 200, 400]$
$n_d = [5, 6, 7]$ |
| SVM | The regularization parameter, $C$
The kernel coefficient for *rbf*, $\gamma$ | $C = [1, 10, 100, 1000, 10{,}000]$
$\gamma = [1 \times 10^{-5}, 1 \times 10^{-4}, 1 \times 10^{-3}, 1 \times 10^{-2}, 1 \times 10^{-1}]$ |
| *Shared hyperparameters for all deep learning models* | | |
| Common | The learning rate, $lr$
The batch size, $bs$
The epoch, $\gamma$ | $lr = [5 \times 10^{-4}, 1 \times 10^{-4}, 5 \times 10^{-5}, 1 \times 10^{-5}]$
$bs = [10, 32, 64, 128]$
$\gamma = 500$ |
| *Specific hyperparameters for each model* | | |
| GCN | The dimension of hidden node features, $h_n$
The dimension of hidden transformer nodes, $h_t$
The dimension of predictor, $h_p$ | $h_n = [64, 128, 256]$
$h_t = [64, 128, 256]$
$h_p = [128, 256, 512]$ |
| GAT | The dimension of hidden node features, $h_n$
The dimension of hidden transformer nodes, $h_t$
The dimension of predictor, $h_p$ | $h_n = [64, 128, 256]$
$h_t = [64, 128, 256]$
$h_p = [128, 256, 512]$ |
| MPNN | The dimension of hidden node features, $h_n$
The dimension of hidden edge features, $h_e$
The dimension of hidden transformer nodes, $h_t$ | $h_n = [64, 128, 256]$
$h_e = [64, 128, 256]$
$h_t = [64, 128, 256]$ |
| AFP | The dimension of hidden node features, $h_n$
The number of iterations for readout, $n_r$ | $h_n = [64, 128, 256]$
$n_r = [1, 2, 3, 4, 5]$ |
| LSTM | *- pretrained* | *- pretrained* |
| Transformer | *- pretrained* | *- pretrained* |
| 1D CNN | The size of convolution kernel, $h_c$
The dimension of hidden features, $h_t$ | $h_c = [4, 8, 10]$
$h_t = [64, 128, 256, 512, 1024]$ |
| IGN | The dimension of hidden features, $h_t$ | $h_t = [64, 128, 256]$ |
| SS-GNN | *- pretrained* | *- pretrained* |

# B    ADDITIONAL RESULTS AND FIGURES

**More dataset features.** Figure 9 illustrates three examples of removed targets and ligands. Figure 14 analyzes the proportion of shared atoms between the AC pairs in the target CHEMBL218 $EC_{50}$ using Maximum Common Substructure (MCS). The average proportion of shared atoms (86.78%) in the identified AC pairs confirms high structural similarity in common substructures.

**Dataset split.** We split the dataset using the Tanimoto similarity of the ECFP. To assess potential bias from ECFP-based data splitting, Figure 16 evaluates ML methods using four molecular descriptors on an alternative MACCS-based split. ECFP maintains superior performance, confirming its inherent descriptive power.

**Protein flexibility.** Using DSDPFlex (Dong et al., 2024), we investigate protein flexibility by allowing flexible side chains for 10 amino acids nearest to the crystal ligand. Figure 17 shows that performance metrics on 8 $K_i$ targets distribute evenly around the $y = x$ line, suggesting comparable effectiveness between fixed and flexible docking approaches.

**Train cross-target models with 3D data.** Table 7 explore the cross-target applicability of 3D models on combined $K_i$ targets under two settings: out-of-distribution (OOD) excluding Protease targets, and in-domain, using all $K_i$ targets. Figure 13 shows multi-target training performs comparable to single-target training, complementing the analysis in § 5.4.

**Combine the 3D information and ECFP features.** To explore the integration of 3D structural information with handcrafted ECFP features, we utilize a 3D model as a feature extractor and combine its output with ECFP descriptors, followed by MLP for affinity prediction (architecture shown Figure 8) The evaluation across ten targets (shown in Table 6) highlights two key findings. First, models incorporating 3D information consistently outperform or match those without 3D information across most targets, achieving notable improvements in overall RMSE and RMSEcliff. Second, the integration of 3D features significantly enhances the model's ability to handle activity cliffs, as evidenced by greater improvements in RMSEcliff (avg. imp. of 5.61%) compared to overall RMSE (avg. imp. of 3.48%).

**Benchmarking the zero-shot ability of more 3D models.** To explore the generalization ability of recent 3D binding affinity prediction models, we evaluate six SOTA methods (PIGNet (Moon et al., 2022), RTMScore (Shen et al., 2022), TANKBind (Lu et al., 2022), DSMBind (Jin et al., 2023), KarmaDock (Zhang et al., 2023b), and EquiScore (Cao et al., 2024)) trained on PDBBind. Figure 15 presents their Pearson correlation on the complete dataset and activity cliff cases across each target. All these methods perform worse on the AC samples, which is consistent with the result of our benchmark. Additionally, these methods show decreased performance compared to the PDBBind test set, with effectiveness correlating with the presence of homologous proteins in the PDBBind training data. For instance, targets with numerous homologous samples in PDBBind demonstrate superior results: CHEMBL2147 $K_i$ achieves a Pearson correlation of 0.688 (DSMBind, PDB ID: 2j2i) with 103 homologous samples, while CHEMBL2971 $K_i$ reaches 0.671 (DSMBind, PDB ID: 4jia) with 61 homologous samples in PDBBind. In contrast, targets lacking homologous proteins in PDBBind (CHEMBL219 $K_i$, CHEMBL228 $K_i$, and CHEMBL233 $K_i$) show very small correlation (DSMBind, Pearson=-0.021, -0.087, and 0.033 respectively).

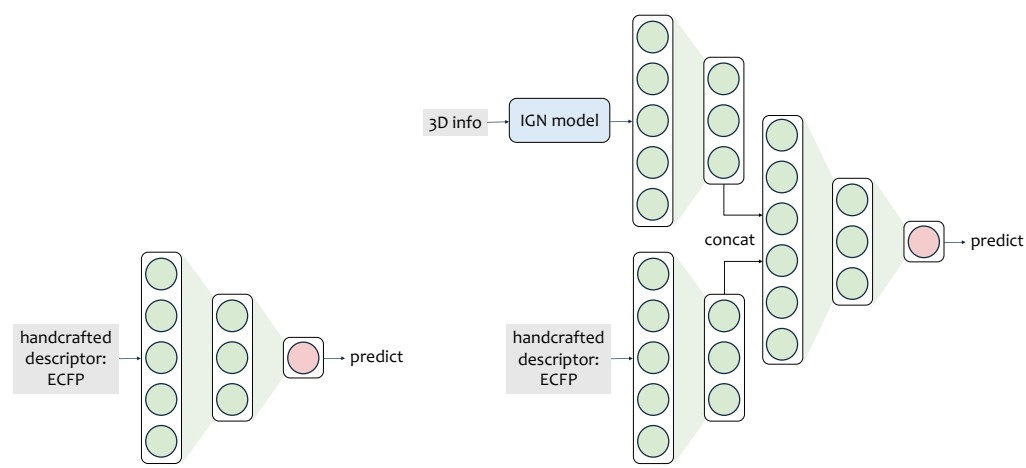

(a) The model illustration of MLP with ECFP descriptor (b) The model illustration of IGN combined with ECFP descriptor

Figure 8: The model illustration of MLP and IGN using the handcrafted molecule descriptor ECFP.

Table 6: The performance of MLP and IGN using the handcrafted molecule descriptor ECFP.

| Model | CHEMBL205 $K_i$ | | CHEMBL214 $K_i$ | | CHEMBL233 $K_i$ | | CHEMBL237 $K_i$ | | CHEMBL264 $K_i$ | |
|---|---|---|---|---|---|---|---|---|---|---|
| | RMSE | RMSE$_{\text{cliff}}$ | RMSE | RMSE$_{\text{cliff}}$ | RMSE | RMSE$_{\text{cliff}}$ | RMSE | RMSE$_{\text{cliff}}$ | RMSE | RMSE$_{\text{cliff}}$ |
| MLP | 0.795 | 0.929 | **0.683** | **0.770** | 0.846 | 0.917 | **0.720** | 0.767 | 0.669 | 0.730 |
| IGN | **0.781** | **0.904** | **0.683** | 0.792 | **0.814** | **0.878** | 0.728 | **0.764** | **0.637** | **0.691** |
| Imp (%) | 1.76 | 2.69 | 0.00 | - | 3.78 | 4.25 | - | 0.39 | 4.78 | 5.34 |

| Model | CHEMBL287 $K_i$ | | CHEMBL1871 $K_i$ | | CHEMBL2047 EC50 | | CHEMBL3979 EC50 | | CHEMBL4203 $K_i$ | |
|---|---|---|---|---|---|---|---|---|---|---|
| | RMSE | RMSE$_{\text{cliff}}$ | RMSE | RMSE$_{\text{cliff}}$ | RMSE | RMSE$_{\text{cliff}}$ | RMSE | RMSE$_{\text{cliff}}$ | RMSE | RMSE$_{\text{cliff}}$ |
| MLP | **0.746** | **0.855** | 0.730 | 0.991 | 0.673 | 0.714 | **0.664** | 0.729 | 0.943 | 0.988 |
| IGN | 0.759 | **0.855** | **0.686** | **0.860** | **0.594** | **0.599** | 0.667 | **0.723** | **0.880** | **0.857** |
| Imp (%) | - | 0.00 | 6.03 | 13.22 | 11.74 | 16.11 | - | 0.82 | 6.68 | 13.26 |

Table 7: The results of Protease in the setting of training with in-domain and out-of-distribution (OOD) targets.

| Model | CHEMBL204 Ki | | CHEMBL244 Ki | | CHEMBL3286 Ki | | CHEMBL4822 Ki | |
|---|---|---|---|---|---|---|---|---|
| | RMSE | RMSE$_{\text{cliff}}$ | RMSE | RMSE$_{\text{cliff}}$ | RMSE | RMSE$_{\text{cliff}}$ | RMSE | RMSE$_{\text{cliff}}$ |
| IGN | 0.873 | 1.027 | 0.891 | 1.006 | 0.724 | 0.829 | 0.751 | 0.778 |
| IGN OOD | 1.612 | 1.788 | 1.647 | 1.643 | 1.183 | 1.149 | 1.153 | 1.197 |

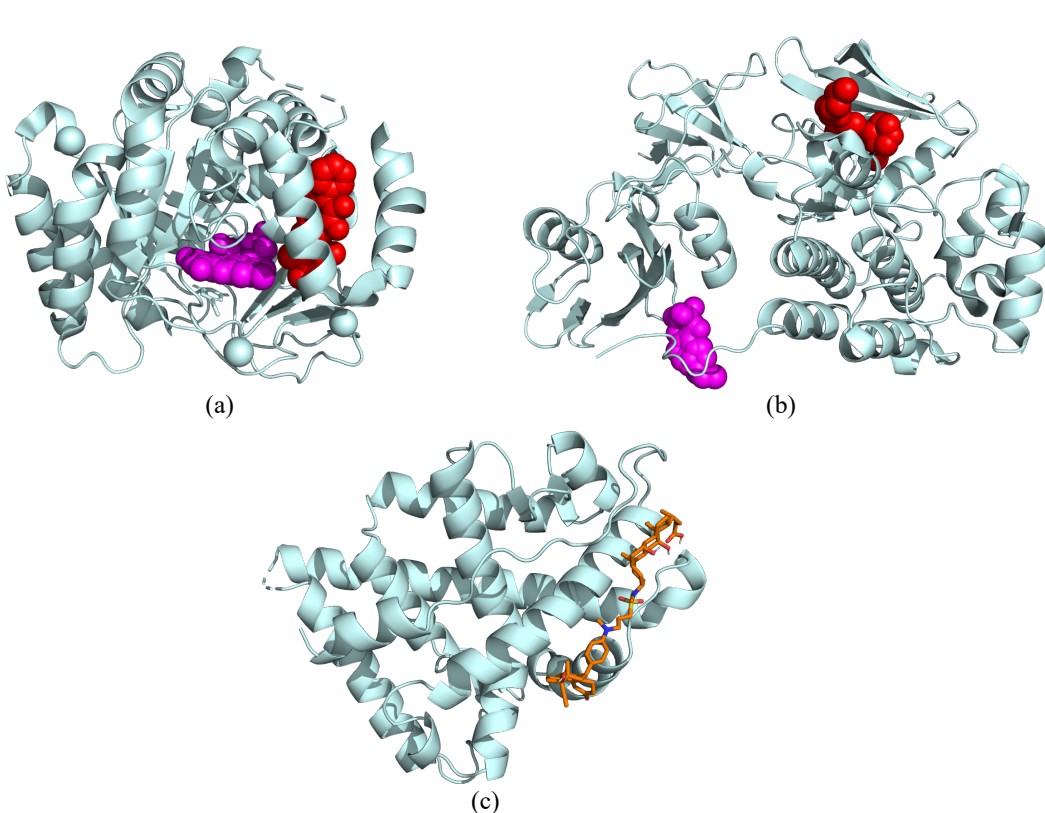

(a)

(b)

(c)

Figure 9: Three examples of the removed targets and ligands. (a) The target structure has two ligand binding sites (PDB: 5mvd). (b) Two structures of the same target have different binding sites (PDB: 2h8h and 1o4j). The two structures are aligned. (c) The ligand docking score is larger than zero (Target: ChEMBL1871, PDB: 2ama, ligand: ChEMBL406027).

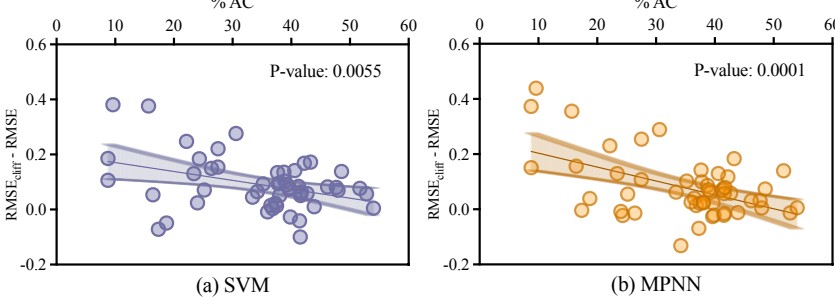

(a) SVM

(b) MPNN

Figure 10: Relationship between the ratio of the AC and $RMSE - RMSE_{cliff}$ of SVM and MPNN.

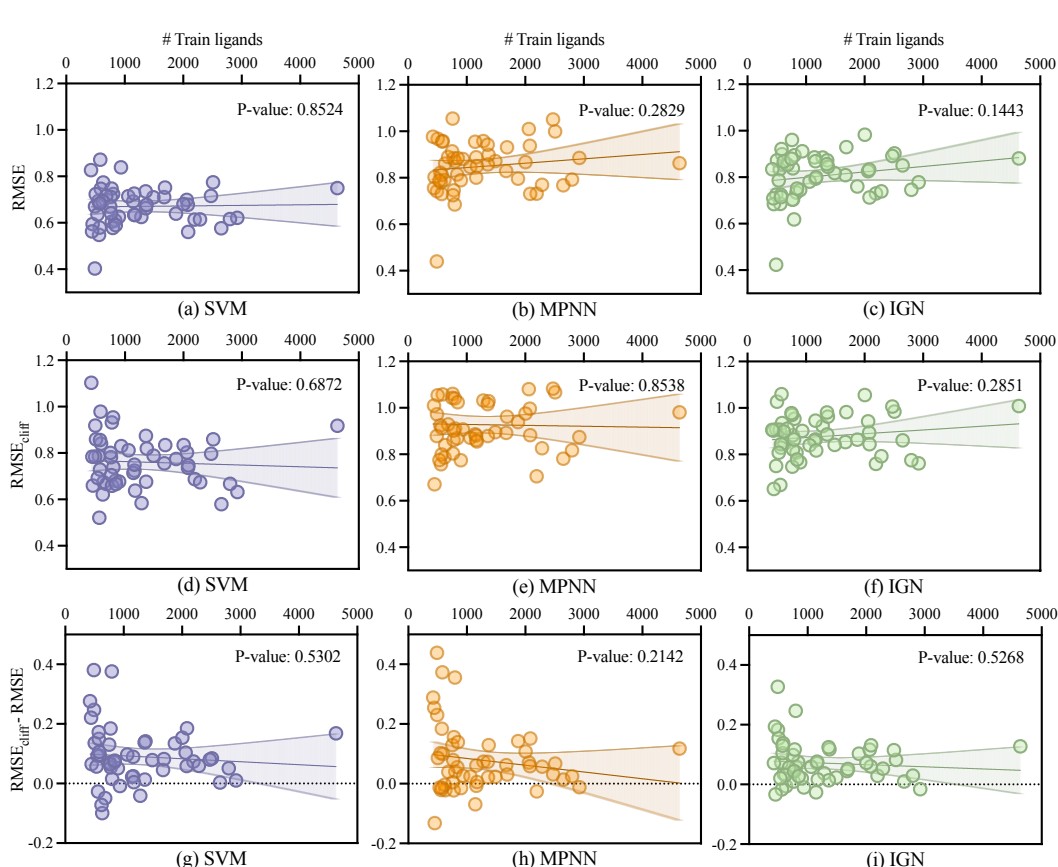

Figure 11: Relationship between the number of training ligands and (a)-(c) RMSE, (d)-(f) RMSE$_{\text{cliff}}$ and (g)-(i) their difference on SVM, MPNN, and IGN.

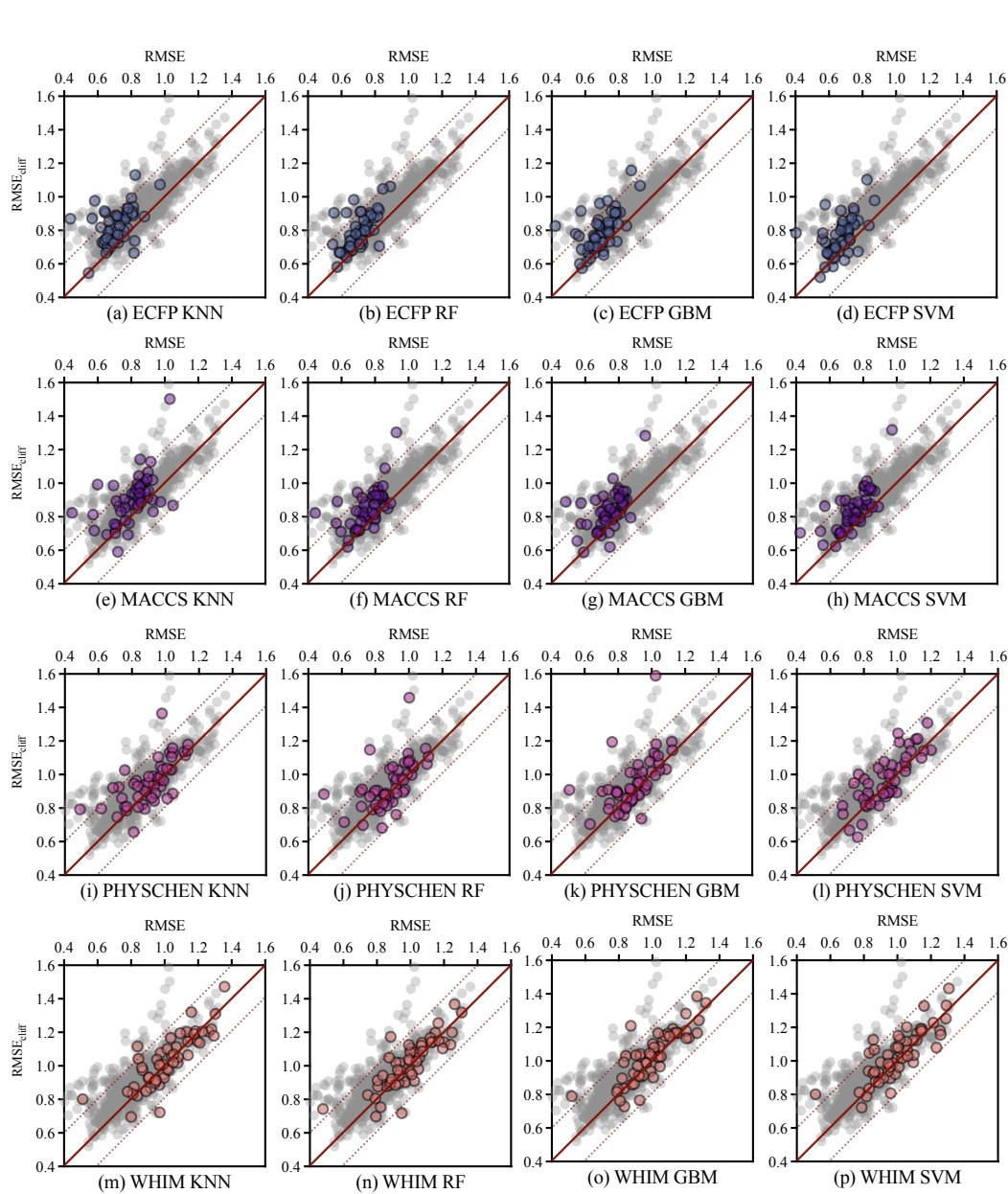

Figure 12: Performance comparison between RMSE and RMSE$_{\text{cliff}}$ for classic ML algorithms across 52 targets.

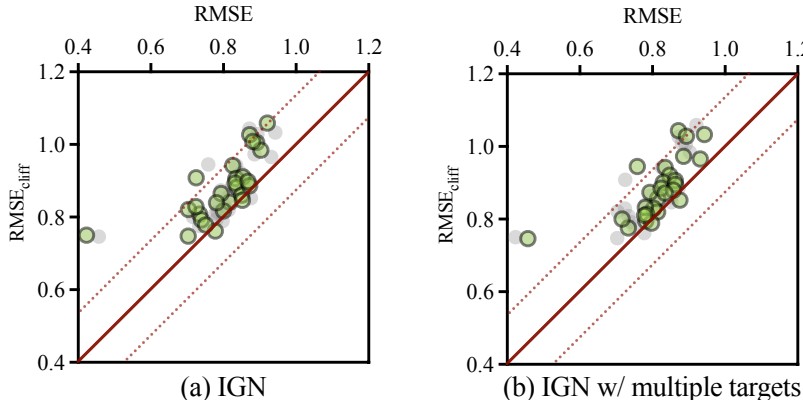

(a) IGN

(b) IGN w/ multiple targets

Figure 13: The results of IGN on all the targets of $K_i$ labels when trained separately on (a) each target or (b) the data of multiple targets.

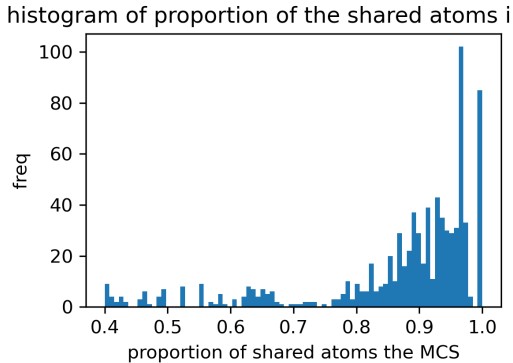

Figure 14: A histogram showing the proportion of the shared atoms between the AC data pairs with Maximum Common Substructure (MCS) in the Target CHEMBL218 $EC_{50}$.

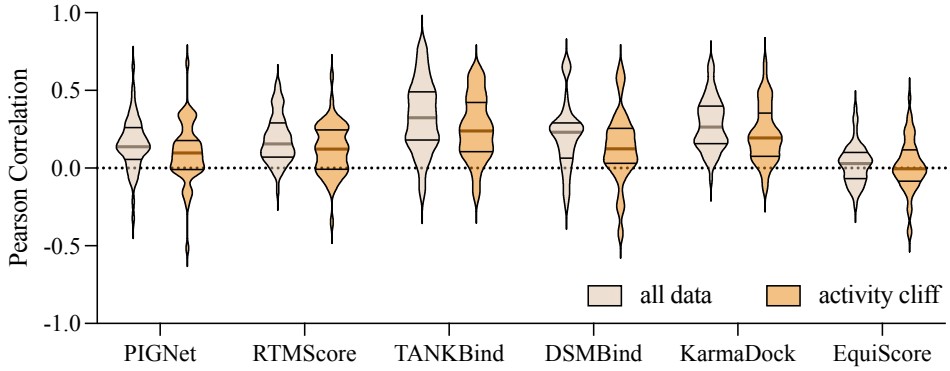

Figure 15: The Pearson and Pearson$_{\text{cliff}}$ evaluated on our DockedAC benchmark across 52 targets using PIGNet (Moon et al., 2022), RTMScore (Shen et al., 2022), TANKBind (Lu et al., 2022), DSMBind (Jin et al., 2023), KarmaDock (Zhang et al., 2023b), and EquiScore (Cao et al., 2024), all of which were trained on general binding affinity datasets.

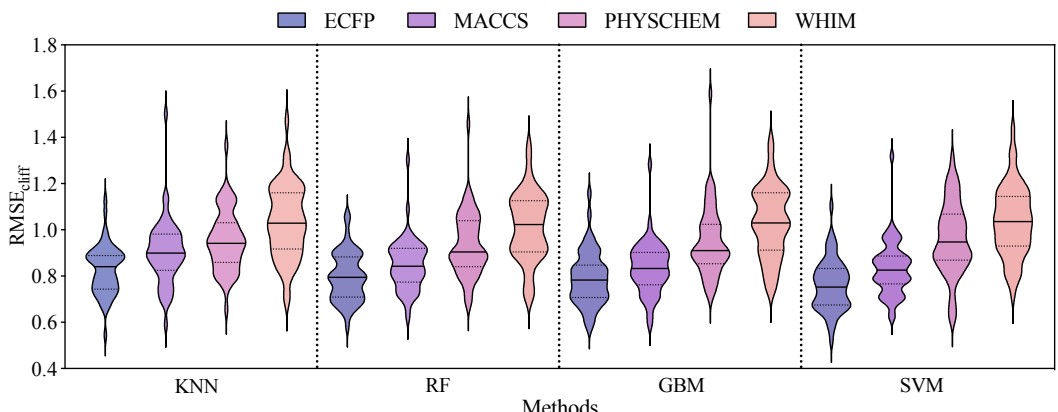

Figure 16: The RMSE$_{cliff}$ evaluated using ML methods under MACCS split across 52 targets.

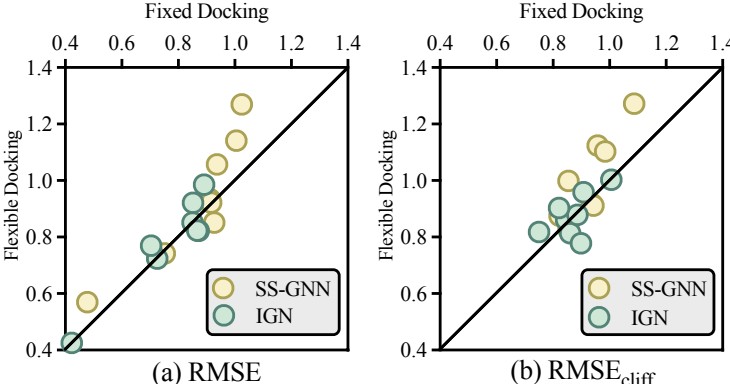

Figure 17: The (a) RMSE and (b) RMSE$_{cliff}$ metric on fixed docking v.s. flexible docking.

