# OpenReview forum: "DockedAC: Empowering Deep Learning Models With 3D Protein-ligand Data For Activity Cliff Analysis"
_ICLR.cc/2025/Conference — Submitted to ICLR 2025_

### Official Review · Reviewer_Z9AJ · 2024-10-28

**Soundness:** 3
**Presentation:** 3
**Contribution:** 3
**Rating:** 6
**Confidence:** 4

**Summary:**

An activity cliff (AC) refers to cases where structurally similar ligands exhibit significantly different activity values due to interactions between protein targets and ligands. Benchmark datasets for ACs have been limited. While existing large datasets consist only of 2D information, they fail to consider 3D interactions. In this context, this work creates a new dataset that adds 3D information which previous AC data could not cover. The purpose and methodology are sound, but there’s a lack of analysis on whether the data classified as activity cliffs genuinely represent real AC relationships. Additionally, there’s no mention of the diversity of conformations generated by docking, and some analyses are not straightforward.

**Strengths:**

**S1**: The authors collected significantly more AC data than previously available high-quality AC datasets[1,2,3] through binding affinity data curation for various protein targets and species.

**S2**: Based on the importance of considering 3D information in target-specific AC prediction, the authors generated pseudo-binding structures using molecular docking, overcoming limitations of 2D-only AC data.

**S3**: The authors benchmarked their dataset using various models, providing many baselines.

## References
[1] Wang, Lingle, et al. "Accurate and reliable prediction of relative ligand binding potency in prospective drug discovery by way of a modern free-energy calculation protocol and force field." Journal of the American Chemical Society 137.7 (2015): 2695-2703.

[2] Schindler, Christina EM, et al. "Large-scale assessment of binding free energy calculations in active drug discovery projects." Journal of Chemical Information and Modeling 60.11 (2020): 5457-5474.

[3] Pecina, A., Fanfrlík, J., Lepšík, M. et al. SQM2.20: Semiempirical quantum-mechanical scoring function yields DFT-quality protein–ligand binding affinity predictions in minutes. Nat Commun 15, 1127 (2024).

**Weaknesses:**

**W1**: It appears that the authors directly followed the definition from existing work. In the definition of activity cliff in Section 3.2, all criteria seem to *implicitly* consider the structural similarity of molecules. Is there any reason not to consider explicit structures such as Maximum Common Substructure (MCS)? If computational complexity is a concern, it would be more impactful to check the distributions of number of shared and different atoms between newly created AC data pairs with MCS.

**W2**: Since the structures are generated through docking, it’s challenging to obtain the correct global minimum structures (as mentioned in the limitations). Providing multiple local minimum structures might be beneficial, but those information are not exist in the current manuscript.

**W3**: Some analyses about dataset is not straightforward. (see Q2-5)

**W4**: Several recent models[1,2,3] aiming versatile predictions for protein-ligand interaction (doing both virtual screening and scoring on acitivity-cliff data) also used well-curated activity cliff data[3,4]. The authors might include these works in related works or further do benchmarks on these models to enlarge the impact of this work.

## References
[1] Moon, Seokhyun, et al. "PIGNet2: a versatile deep learning-based protein–ligand interaction prediction model for binding affinity scoring and virtual screening." Digital Discovery 3.2 (2024): 287-299.

[2] Shen, Chao, et al. "A generalized protein–ligand scoring framework with balanced scoring, docking, ranking and screening powers." Chemical Science 14.30 (2023): 8129-8146.

[3] Cao, Duanhua, et al. "Generic protein–ligand interaction scoring by integrating physical prior knowledge and data augmentation modelling." Nature Machine Intelligence (2024): 1-13.

[4] Pecina, A., Fanfrlík, J., Lepšík, M. et al. SQM2.20: Semiempirical quantum-mechanical scoring function yields DFT-quality protein–ligand binding affinity predictions in minutes. Nat Commun 15, 1127 (2024).

[5] Schindler, Christina EM, et al. "Large-scale assessment of binding free energy calculations in active drug discovery projects." Journal of Chemical Information and Modeling 60.11 (2020): 5457-5474.

**Questions:**

**Q1**: Sections 3.2 and 3.5 seem identical to previous work[1]. Shouldn’t this be acknowledged or cited appropriately?

**Q2**: In Section 5.1, when correlating RMSE with RMSE-cliff, is there a potential for bias introduced by comparing AC data with the entire dataset (which includes both AC and non-AC data)?

**Q3**: Conceptually, using 3D information should improve AC prediction. However, deriving this conclusion in section 5.1 from the results comparing IGN[2] and other 2D models seems to lack depth in interpretation. It would be better to compare IGN with a 2D version of IGN to minimize bias.

**Q4**: In Section 5.3, is the sample size too small for calculating the p-value? Additionally, in Table 3, the percentage of AC is at most 0.6%. Why did the authors choose to show the correlation with a maximum AC data percentage of only 0.6%?

**Q5**: In Section 5.4, ECFP shows the best performance. Could this be because ECFP was used in the criteria when dividing the data? Is there a potential ECFP bias? How do the results look if the authors use the other fingerprints for dividing data instead of ECFP?

## References
[1] Van Tilborg, Derek, Alisa Alenicheva, and Francesca Grisoni. "Exposing the limitations of molecular machine learning with activity cliffs." Journal of chemical information and modeling 62.23 (2022): 5938-5951.

[2] Jiang, Dejun, et al. "Interactiongraphnet: A novel and efficient deep graph representation learning framework for accurate protein–ligand interaction predictions." Journal of medicinal chemistry 64.24 (2021): 18209-18232.

---

> ### Author Response · Authors · 2024-11-25
> **Thank you for your feedback! (1/2)**
>
> Thank you for your detailed feedback and insightful suggestion regarding the structural similarity assessment in the activity cliff definition. We will address your comments in the responses below.
>
> **[W1. Maximum Common Substructure]**
>
> We appreciate the suggestion to incorporate explicit structural comparison methods like Maximum Common Substructure (MCS), as it could provide valuable insights.
>
> While computational complexity is a consideration, we have computed the number of shared atoms between the newly generated AC pairs, as shown in the newly added Appendix Figure 14 for one of the targets. The average proportion of the shared atoms is 86.78%,  which shows that the identified AC pairs have a high similarity in terms of common substructures.
>
> We will include this analysis in the revised manuscript. This addition will strengthen the rigour of our generated AC pairs and provide deeper insights into the structural aspects of activity cliffs.
>
>
> **[W2. multiple local minimum]**
>
> Thank you for highlighting this important question. We agree that considering multiple local minima would provide a more comprehensive view of potential binding modes.
>
> To further enrich our dataset, we will generate top-N scoring poses (N=5-10) and calculate RMSD between different poses. We will include multiple poses in the dataset and provide pose scoring information.
>
> The reason for that we currently only use the top-1 binding mode in the dataset is because, there are no good deep learning models designed to process multiple input conformations. This would be a very important direction for future work, as the binding affinity is the result of a binding/unbinding dynamic process. We will add this discussion to the revised paper.
>
>
> **[W3. analyses about dataset]**
>
> > **Q2**: In Section 5.1, when correlating RMSE with RMSE-cliff, is there a potential for bias introduced by comparing AC data with the entire dataset (which includes both AC and non-AC data)?
>
> You are right that RMSE includes both AC and non-AC data and RMSE-cliff is calculated from only the AC data.
>
> We compare RMSE-cliff with RMSE to show that the prediction with AC data is a more difficult task, which can be seen from the result that most of the targets are lying above the line RMSE = RMSE-cliff (in Figure 4). It means that the average RMSE on the AC samples is higher than the average RMSE on all the samples, i.e.,
>
> $$ RMSE = \frac{\\# AC * RMSE_{cliff} + \\# {nonAC} * RMSE_{nonAC}}{\\#AC+ \\#nonAC} < RMSE_{cliff}. $$
>
> $\\#$
> From the above, we can see that $RMSE_{cliff} > RMSE_{nonAC}$. Please let us know if you have any further questions.
>
>
> > **Q3**: Conceptually, using 3D information should improve AC prediction. However, deriving this conclusion in section 5.1 from the results comparing IGN[2] and other 2D models seems to lack depth in interpretation. It would be better to compare IGN with a 2D version of IGN to minimize bias.
>
> The inputs of the 2D version and the 3D version are not the same. The 3D model has the pocket and ligand 3D structure as additional inputs. So it is not very straightforward to compare the two cases.
> In our paper, we do such a comparison in two different aspects. The first is to compare IGN or SS-GNN to other 2D graph neural network models, as shown in section 5.1. We chose IGN and SS-GNN because they are based on similar graph neural network architectures. For example, SS-GNN uses GIN as the backbone and uses the 3D information to enhance the edge features in the GIN. In the second aspect, we use a 3D model as a feature extractor and combine the 3D features with the handcrafted 2D features and employ MLP for affinity prediction, as shown in the Appendix Figure 8 and Table 6. The results from the second aspect also show that the features from 3D data can enhance the AC prediction.
>
>
> > **Q4**: In Section 5.3, is the sample size too small for calculating the p-value? Additionally, in Table 3, the percentage of AC is at most 0.6%. Why did the authors choose to show the correlation with a maximum AC data percentage of only 0.6%?
>
> Thanks for your careful examination of our paper. 0.6 should be the proportion instead of the percentage. So the percentage is 60%. We have corrected the caption and figure label.

---

> > ### Author Response · Authors · 2024-11-25
> > **Thank you for your feedback! (2/2)**
> >
> > > **Q5**: In Section 5.4, ECFP shows the best performance. Could this be because ECFP was used in the criteria when dividing the data? Is there a potential ECFP bias? How do the results look if the authors use the other fingerprints for dividing data instead of ECFP?
> >
> > The observation that ECFP's superior performance might be influenced by its use in data division criteria is an interesting point. Thanks for pointing it out.
> >
> > We split the data again using the similarity computed with the MACCS fingerprint. The result does not change and ECFP still has the best performance. ECFPs are designed to capture molecular features relevant to molecular activity and are particularly effective for similarity searching and structure-activity relationship (SAR) studies. ECFP captures the environment around each atom iteratively and builds a detailed local structural context.
> >
> > |Method | ECFP | MACCS | PHYSCHEM | WHIM |
> > |-|-|-|-|-|
> > |**KNN**|0.8093|0.8727|0.9588|1.0388|
> > |**RF**|0.7653|0.8248|0.9277|1.0179|
> > |**GBM**|0.7608|0.8033|0.9354|1.0093|
> > |**SVM**|0.7430|0.8189|0.9529|1.0300|
> >
> > We also present the full results of each target in the revised manuscript (see Appendix Figure 16).
> >
> > **[Questions]**
> >
> > >**Q1**: Sections 3.2 and 3.5 seem identical to previous work[1]. Shouldn’t this be acknowledged or cited appropriately?
> >
> > Thank you for bringing this important issue to our attention regarding the missing reference in Sections 3.2 and 3.5 to previous work. We apologize for any oversight in proper attribution. We will add explicit citations to the related work in both sections.
> >
> > ----
> >
> > We hope that we have properly addressed your concerns. Please let us know if you have any further insights or questions answering which would make you feel more positive about our paper. Thank you again for your time and valuable feedback.

---

> > > ### Comment · Reviewer_Z9AJ · 2024-11-27
> > >
> > > Thanks for the authors for providing extensive additional results. My concerns are mostly resolved, so I'll raise my score from 5 to 6.

---

> > > > ### Author Response · Authors · 2024-11-28
> > > > **Thanks for your reply!**
> > > >
> > > > We sincerely appreciate your thorough review and insightful feedback. Should you have any additional questions or concerns, we would be pleased to address them.

---

### Official Review · Reviewer_sP4z · 2024-10-30

**Soundness:** 2
**Presentation:** 3
**Contribution:** 2
**Rating:** 6
**Confidence:** 3

**Summary:**

This work introduces the new benchmark dataset including Activity Cliffs (AC), which holds significant importance in quantitative structure-activity relationship (QSAR) analysis.
Authors collect the bioactivity data from ChEMBL, refine the collected molecules, find the AC relationships in the dataset, and construct 3D conformer dataset using docking.
Finally, authors investigate the regression performance of various ML and DL methods and demonstrate that the 3D spatial information helps to distinguish ACs.

**Strengths:**

1. Authors include various ML, DL methods for benchmarking.
2. The curation process is clear and well-organized to understand.
3. Unlike previous datasets, DockedAC includes 3D conformers which play crucial roles in protein-ligand interaction.
4. They perform an ablation study to investigate the 3D conformational information. (Appendix B)

**Weaknesses:**

**Issue 1.**
I believe that a benchmark set like DockedAC, which focuses on metrics relevant to the application domain, its value is evaluated by how effectively it can assess the performance of different methodologies used in the field (e.g., PoseCheck [1], DUD-E [2]).
In this light, I think this AC-focused benchmark can be used as a rigorous and meaningful test set for various 3D protein-ligand interaction prediction model, which have been in development for a long time.

Therefore, I wonder why authors do not include evaluations for popular 3D binding affinity prediction models (e.g., PIGNet [3], RTMScore [4], TANKBind [5], DSMBind [6], KarmaDock [7]).
Similar to how DUD-E serves as a virtual screening benchmark, the authors could evaluate these models by using the entire collected dataset as a test set.
Although the authors included IGN and SS-GNN, which are 3D binding affinity prediction models, they trained these neural networks on their own training set.
I suggest evaluating the activity cliff prediction performance of some of state-of-the-art models [3-7] trained on general binding affinity dataset (e.g., PDBbind) using the DockedAC benchmark set.

**Issue 2.**
The 3D conformer sets are generated through a single GPU-accelerated docking using AutoDock Vina scoring. Since the binding conformations are crucial for identifying protein-ligand interactions, and I concerned that the performance of the 3D DL model may be constrained by the quality of the conformers. It would be beneficial to include the diverse conformers computed by various docking tools employing different scoring functions, such as conventional docking (Smina, AutoDock Vina, Quick Vina, GLIDE) or deep learning-based docking (KarmaDock, DiffDock).

---
**Reference**

1. Harris, Charles, et al. "Posecheck: Generative models for 3d structure-based drug design produce unrealistic poses." NeurIPS 2023 Generative AI and Biology (GenBio) Workshop. 2023.
2. Mysinger, Michael M., et al. "Directory of useful decoys, enhanced (DUD-E): better ligands and decoys for better benchmarking." Journal of medicinal chemistry 55.14 (2012): 6582-6594.
3. Moon, Seokhyun, et al. "PIGNet: a physics-informed deep learning model toward generalized drug–target interaction predictions." Chemical Science 13.13 (2022): 3661-3673.
4. Shen, Chao, et al. "Boosting protein–ligand binding pose prediction and virtual screening based on residue–atom distance likelihood potential and graph transformer." Journal of Medicinal Chemistry 65.15 (2022): 10691-10706.
5. Lu, Wei, et al. "Tankbind: Trigonometry-aware neural networks for drug-protein binding structure prediction." Advances in neural information processing systems 35 (2022): 7236-7249.
6. Jin, Wengong, Caroline Uhler, and Nir Hacohen. "SE (3) denoising score matching for unsupervised binding energy prediction and nanobody design." NeurIPS 2023 Generative AI and Biology (GenBio) Workshop.
7. Zhang, Xujun, et al. "Efficient and accurate large library ligand docking with KarmaDock." Nature Computational Science 3.9 (2023): 789-804.

**Questions:**

1. Could you please clarify whether the 3D DL methods are trained on the entire training set using a single model that is generalized to pockets (target-conditioned) or are they trained for each target pocket (target-specific)? Additionally, what are the target DL methods of this benchmark? Neural network architecture (e.g., EGNN or MPNN) or trained 3D DL models which is generalized to different proteins? (e.g., PIGNet, KarmaDock.)
3. I wonder why empirical scoring function of AutoDock Vina is not included as the baseline.
4. As I understand it, the ligands are docked to the single rigid protein binding site structure. To reflect the protein’s flexibility, it would be helpful to consider multiple protein structures, i.e., run docking with multiple protein structures for each ligand.

---

> ### Author Response · Authors · 2024-11-25
> **Thank you for your feedback! (1/2)**
>
> Thank you for your insightful feedback. Your suggestions will help strengthen our benchmark's utility and comprehensiveness.
>
> **Issue 1. [Regarding the evaluation of existing 3D binding affinity prediction models]**
>
> We appreciate your feedback regarding the evaluation scope of our benchmark. Our initial inclusion of IGN and SS-GNN was motivated by their graph-based architectures, enabling direct comparison with 2D GNN models (GCN, GIN) to demonstrate the impact of 3D structural information.
>
> Following your suggestions, we have now extended the evaluation of PIGNet [1], RTMScore [2], TANKBind [3], DSMBind [4], KarmaDock [5] and EquiScore [6] on our collected dataset and the averaged results across the targets are in the following table (measured by Pearson correlation). We also include the full results of each target in the revised manuscript (Appendix Figure 15).
>
> In general, the performance of these methods is not as good as that of the PDBbind test set. In the targets that have many homologous proteins in the PDBBind training set, the model tends to work better. For example, CHEMBL2147_Ki (DSMBind Pearson=0.688, pdb id 2j2i) has 103 homologous samples in the PDBBind dataset. CHEMBL2971_Ki (DSMBind Pearson=0.671, pdb id 4jia) has 61 homologous samples. While the proteins of CHEMBL219_Ki, CHEMBL228_Ki and CHEMBL233_Ki can not be found in the PDBBind dataset, DSMBind Pearson=-0.021, -0.087 and 0.033 respectively.
>
> ||PIGNet| RTMScore|TANKBind| DSMBind |KarmaDock| EquiScore |
> |-|-|-|-|-|-|-|
> |Pearson|0.1560 |0.1854 |0.3416  |0.2050    |0.2831 | 0.0280 |
> |Pearson_cliff|0.0964 |0.1208  |0.2569 |0.1283  |0.2196 |0.0127 |
>
>
> In addition, one of the key advantages of 3D approaches is their potential for cross-target applicability. To explore this, we retrain one of the 3D methods (IGN) on multiple targets to assess the cross-target performance. Though we cannot get the full results due to the limited time, from the current results, there are two observations. First, when trained with multiple targets, the performance is comparable to training from scratch on the specific target. Second, when the protein type (Protease) of the testing targets is not in the training targets, the performance drops from 0.9 to 1.4. This is consistent with previous studies [7][8]. When the targets are split by the protein sequence identity of 30%, the RMSE increases from 1.27 (random split) to 1.429 (30% identity split) (See Appendix Table 7 in the revised manuscript).
>
> [1] Moon, Seokhyun, et al. "PIGNet: a physics-informed deep learning model toward generalized drug–target interaction predictions." Chemical Science 13.13 (2022): 3661-3673.
>
> [2] Shen, Chao, et al. "Boosting protein–ligand binding pose prediction and virtual screening based on residue–atom distance likelihood potential and graph transformer." Journal of Medicinal Chemistry 65.15 (2022): 10691-10706.
>
> [3] Lu, Wei, et al. "Tankbind: Trigonometry-aware neural networks for drug-protein binding structure prediction." Advances in neural information processing systems 35 (2022): 7236-7249.
>
> [4] Jin, Wengong, Caroline Uhler, and Nir Hacohen. "SE (3) denoising score matching for unsupervised binding energy prediction and nanobody design." NeurIPS 2023 Generative AI and Biology (GenBio) Workshop.
>
> [5] Zhang, Xujun, et al. "Efficient and accurate large library ligand docking with KarmaDock." Nature Computational Science 3.9 (2023): 789-804.
>
> [6] Cao, Duanhua, et al. "Generic protein–ligand interaction scoring by integrating physical prior knowledge and data augmentation modelling." Nature Machine Intelligence (2024): 1-13.
>
> [7] SS-GNN: A Simple-Structured Graph Neural Network for Affinity Prediction. ACS Omega. 2023, 8(25): 22496–22507.
>
> [8] Pre-Training of Equivariant Graph Matching Networks with Conformation Flexibility for Drug Binding. Advanced Science, 2022, 9(33): 2203796

---

> > ### Author Response · Authors · 2024-11-25
> > **Thank you for your feedback! (2/2)**
> >
> > **Issue 2. [Regarding conformer generation diversity]**
> >
> > We acknowledge the limitation of using a single docking tool, and thank you for the suggestion of using more docking tools.
> >
> > To further enrich our dataset, we have included the docking pose generated by KarmaDock in the dataset, which contains three different versions, i.e., KarmaDock, KarmaDock+FF (corrected with force field) and KarmaDock+Align (corrected with alignment).
> >
> > Besides, we make the top 10 residues that are closest to the reference ligand have flexible side chains, and use DSDPflex [1] to generate more docking conformations. The newly generated data will be updated in the url link.
> >
> > [1] Dong C, Huang Y P, Lin X, et al. DSDPFlex: Flexible-Receptor Docking with GPU Acceleration[J]. Journal of Chemical Information and Modeling, 2024.
> >
> >
> > **Question 1. [target-specific and target-conditioned training]**
> >
> > Thank you for raising this important question. The results in the paper are all from the models trained for each target separately. This is for comparison to the 2D GNN methods. We have modified the paper and made this clearer.
> >
> > Further, we use the targets with Ki labels and train a single model that generalizes across targets (target-conditioned). The results are shown in Appendix Figure 13 in the revised manuscript. The performance is comparable to training from scratch on a specific target.
> >
> > The main target DL methods for benchmarking is about the neural network architectures or deep learning methods that are able to model the 3D complex structures. From our experiments, we observe that some traditional machine learning approaches with designed features achieve competitive performances, indicating significant opportunities for advancing AC prediction based on 3D deep learning methods.
> >
> >
> > **Question 2. [AutoDock Vina score]**
> >
> > We have computed the correlation between experimental activity labels and docking scores, finding a very weak correlation (approximately 0.1). Given this limited predictive value, we opted not to include these results in the paper.
> >
> >
> > **Question 3. [protein’s flexibility]**
> >
> > Thank you for highlighting the flexibility of proteins. We have used DSDPflex for docking with flexible side chains. We choose the top 10 closest amino acids to the crystal ligand and make their side chains flexible. And we record the top-5 docking conformations for each ligand.
> >
> > This semi-flexible docking approach generates a more diverse set of 3D structures. However, it also brings the challenge of developing models that can effectively handle multiple structures as inputs.
> >
> > ----
> >
> > We hope that we have properly addressed your concerns. Please let us know if you have any further insights or questions answering which would make you feel more positive about our paper. Thank you again for your time and valuable feedback.

---

> ### Comment · Reviewer_sP4z · 2024-11-26
> **Official Comment from Author**
>
> Thank you for detailed response. I'll increase the score.

---

> > ### Author Response · Authors · 2024-11-28
> > **Thanks for your reply!**
> >
> > Thanks for your constructive comments and for raising your score!  If you have any further questions or concerns, please don't hesitate to let us know.

---

### Official Review · Reviewer_mD8Y · 2024-10-30

**Soundness:** 2
**Presentation:** 3
**Contribution:** 4
**Rating:** 6
**Confidence:** 3

**Summary:**

**Summary:** The authors present DockedAC for comprehensive machine learning of molecular activity cliffs.

**Recommendation:** I am currently recommending a weak (yet optimistic) reject.

**Rationale behind Recommendation:** If the authors can clarify their dataset splitting approach and potentially benchmark one or two more recent 3D GNN/transformer architectures, I will consider raising my score.

**Strengths:**

- The authors' experiments and conclusions are thorough and informative. I appreciate the correlation studies for the insights they provide into the potential of 3D GNNs in this problem domain.
- The authors' benchmark is well-constructed overall.
- The authors' data and benchmarking code is thoroughly documented.

**Weaknesses:**

- The authors need to clarify the splitting of their dataset to ensure proper benchmarking (see "Questions" below).
- The authors' chosen 3D GNNs (or 3D/equivariant transformers, which in my view may be most interesting to explore for this problem) are not up-to-date. For example, new models such as the Equiformer v2 architecture [1] could be included as well.
- The authors should consider performing flexible docking with conventional docking algorithms rather than "fixed-protein" docking, since the amino acid side chain rotamer states in a crystal protein structure may be conducive to only the ligand against which the protein was originally bound [2]. I understand this may be computationally expensive, but perhaps the authors can perform small-scale experiments at some point to see if this makes a difference in the final benchmarking results.

**References:**

[1] Liao, Yi-Lun, et al. "EquiformerV2: Improved Equivariant Transformer for Scaling to Higher-Degree Representations." The Twelfth International Conference on Learning Representations.

[2] Wankowicz, Stephanie A., et al. "Ligand binding remodels protein side-chain conformational heterogeneity." Elife 11 (2022): e74114.

**Questions:**

- On line 198, how is a canonical SMILES string determined for each pair of ligands?
- On line 213, when are two binding sites considered the same? Do you use a distance-based metric here? Please explain in more detail.
- On line 227, what does "docking results are reviewed" mean? Do you simply mean that you reject ligands that achieved a docking score greater than or equal to zero? Or did you manually apply some other heuristic to reject ligands at this stage?
- Regarding line 234, could the authors provide more detail about why they split the dataset this way? I ask because it's not clear to me how this splitting strategy would prevent models from overfitting to popular protein targets such as GPCRs if they are widely represented in the dataset. Perhaps I'm misunderstanding how the authors trained their baseline methods on this dataset (e.g., maybe they only trained on one protein at a time, an approach which seems difficult to scale), but it's important for the authors to more clearly explain their design and rationale for this proposed dataset splitting strategy (since their results suggest 3D GNNs perform well in certain contexts, where they might be overfitting to the crystal protein structures in the dataset).
- More of a suggestion: Drawing a downward arrow next to 5.4 nM in Figure 1 could inform readers unfamiliar with bioactivity measurements that a lower quantity is better in this context.

---

> ### Author Response · Authors · 2024-11-25
> **Thank you for your feedback! (1/2)**
>
> Thank you for your careful review and insightful suggestions. We will try to address each point to clarify and improve our manuscript.
>
> **Weakness**
>
> >The authors need to clarify the splitting of their dataset to ensure proper benchmarking
>
> > **Question**: Regarding line 234, could the authors provide more detail about why they split the dataset this way?
>
> Data splitting is an important point for proper benchmarking. In the main paper, we train on one protein at a time, and benchmark the performance for each target. The reason is to compare with the 2D methods, which do not have protein information and can not be trained across proteins.
>
> Besides performance improvement, another advantage of using 3D information is the potential for cross-target applicability. To explore this question and enrich our benchmark results, we combined the targets of the Ki as labels together and trained the IGN model again.
>
> We use the following two settings:
> - out-of-distribution (OOD): excluding one type of target for training (Protease)
> - in-domain: using all the targets of Ki for training
>
> We show the results of testing on the four Protease targets (Appendix Table 7). It is easy to see that when the protein type (Protease) of the testing targets is not in the training targets, the performance (avg. RMSEcliff) drops from 0.9 to 1.4. The results of testing on all the targets are also reported (Appendix Figure 13). From the current results, when trained with multiple targets, the performance is comparable to training from scratch on a specific target. The current results indicate that there is still a long way to go to fully exploit the multi-target 3D data and make the model generalize to new targets.
>
> We have made it clear about the data splitting and model training, and added the new results to the maniscript.
>
> >The authors' chosen 3D GNNs (or 3D/equivariant transformers, which in my view may be most interesting to explore for this problem) are not up-to-date. For example, new models such as the Equiformer v2 architecture [1] could be included as well.
>
> We fully agree that incorporating more new models would significantly enhance the benchmark results.
>
> The main reason for using IGN and SS-GNN for benchmarking is that these two methods are based on graph neural networks and their results are more comparable to 2D GNN models like GCN or GIN.
>
> The Equiformer v2 architecture [1] is for property prediction of a single molecule. It is not straightforward to extend it for affinity prediction from a pocket-ligand complex. But following your advice and suggestions from other reviewers, we evaluate the performance of several other more recent models for binding affinity prediction, including PIGNet [2], RTMScore [3], TANKBind [4], DSMBind [5], KarmaDock [6] and EquiScore [7].
>
> All these methods have model weights trained with the PDBbind dataset and we test them on our collected dataset. The averaged results across the targets are in the following table (measured by Pearson correlation). The full results across 52 targets are presented in the revised manuscript (Appendix Figure 15).
>
> ||PIGNet| RTMScore|TANKBind| DSMBind |KarmaDock| EquiScore |
> |-|-|-|-|-|-|-|
> |Pearson|0.1560 |0.1854 |0.3416  |0.2050    |0.2831 | 0.0280 |
> |Pearson_cliff|0.0964 |0.1208  |0.2569 |0.1283  |0.2196 |0.0127 |
>
>
> Due to the time limit, we re-train the model EquiScore [7] with our dataset and compare with IGN and SS-GNN. The performance of EquiScore is slightly better than SS-GNN, but worse than IGN.
>
> ||IGN| SS-GNN|EquiScore|
> |-|-|-|-|
> |Pearson        |0.7405|0.6721|0.6797|
> |Pearson_cliff |0.6534|0.5813|0.5986|
>
> [1] Liao, Yi-Lun, et al. "EquiformerV2: Improved Equivariant Transformer for Scaling to Higher-Degree Representations." The Twelfth International Conference on Learning Representations.
>
> [2] Moon, Seokhyun, et al. "PIGNet: a physics-informed deep learning model toward generalized drug–target interaction predictions." Chemical Science 13.13 (2022): 3661-3673.
>
> [3] Shen, Chao, et al. "Boosting protein–ligand binding pose prediction and virtual screening based on residue–atom distance likelihood potential and graph transformer." Journal of Medicinal Chemistry 65.15 (2022): 10691-10706.
>
> [4] Lu, Wei, et al. "Tankbind: Trigonometry-aware neural networks for drug-protein binding structure prediction." Advances in neural information processing systems 35 (2022): 7236-7249.
>
> [5] Jin, Wengong, Caroline Uhler, and Nir Hacohen. "SE (3) denoising score matching for unsupervised binding energy prediction and nanobody design." NeurIPS 2023 Generative AI and Biology (GenBio) Workshop.
>
> [6] Zhang, Xujun, et al. "Efficient and accurate large library ligand docking with KarmaDock." Nature Computational Science 3.9 (2023): 789-804.
>
> [7] Cao, Duanhua, et al. "Generic protein–ligand interaction scoring by integrating physical prior knowledge and data augmentation modelling." Nature Machine Intelligence (2024): 1-13.

---

> > ### Author Response · Authors · 2024-11-25
> > **Thank you for your feedback! (2/2)**
> >
> > >The authors should consider performing flexible docking with conventional docking algorithms rather than "fixed-protein" docking, since the amino acid side chain rotamer states in a crystal protein structure may be conducive to only the ligand against which the protein was originally bound [2]. I understand this may be computationally expensive, but perhaps the authors can perform small-scale experiments at some point to see if this makes a difference in the final benchmarking results.
> >
> > Thank you for this valuable suggestion regarding flexible docking. Protein flexibility could significantly impact binding mode prediction.
> >
> > We have tried to use the latest AutoDock Vina (v1.2.5) for flexible docking. We find that it takes a very long time (more than one hour) to dock one ligand if all the residues in the pocket have flexible side chains. So we make the 10 closest residues to the reference ligand have flexible side chains, and use DSDPflex [1] to generate more docking conformations. To further enrich our dataset, we record top-5 scoring poses. The newly generated data will first be added to our dataset. Due to the time limit, we will then run experiments on the targets with Ki labels to see the effects.
> >
> >
> > [1] Dong C, Huang Y P, Lin X, et al. DSDPFlex: Flexible-Receptor Docking with GPU Acceleration[J]. Journal of Chemical Information and Modeling, 2024.
> >
> >
> > **Questions**
> >
> > >On line 198, how is a canonical SMILES string determined for each pair of ligands?
> >
> > The canonical SMILES are from ChEMBL.
> >
> > >On line 213, when are two binding sites considered the same? Do you use a distance-based metric here? Please explain in more detail.
> >
> > Yes, a distance-based metric is used. In particular, we use two metrics, (a) the distance between the centers of the ligands and (b) the minimum distance between the atoms of the two ligands. If the distance between the centers is less than 4Å, then the two binding sites are the same. There are also cases the ligands differ a lot in size. So if the minimum distance between the atoms of the two ligands is less than 0.5Å, we also consider the two binding sites to be the same.
> >
> >
> > >On line 227, what does "docking results are reviewed" mean? Do you simply mean that you reject ligands that achieved a docking score greater than or equal to zero? Or did you manually apply some other heuristic to reject ligands at this stage?
> >
> > Currently, we use the docking score as a criterion to reject the ligands. And the ligand is also rejected if the center of the docked ligands is too far from the center of the binding site (>4Å). This may be alleviated by recording multiple docking conformations, such as top-5 instead of top-1.
> >
> > >More of a suggestion: Drawing a downward arrow next to 5.4 nM in Figure 1 could inform readers unfamiliar with bioactivity measurements that a lower quantity is better in this context.
> >
> > Thanks for your thoughtful suggestion. We have modified the figure.
> >
> >
> > ----
> >
> > We hope that we have properly addressed your concerns. Please let us know if you have any further insights or questions answering which would make you feel more positive about our paper. Thank you again for your time and valuable feedback.

---

> > > ### Comment · Reviewer_mD8Y · 2024-11-25
> > > **Response to author rebuttal**
> > >
> > > Thanks to the authors for their detailed and direct responses to my initial concerns. I believe they have addressed them adequately. As such, I am comfortable raising my score from a weak reject (5) to a weak accept (6), as I believe this paper offers the community a timely new benchmark dataset for an upcoming application area (e.g., dynamic affinity prediction).

---

> > > > ### Author Response · Authors · 2024-11-28
> > > > **Thanks for your reply！**
> > > >
> > > > Thanks for acknowledging our response. We are encouraged by your positive comments. If you have any further questions or concerns, please don't hesitate to let us know.

---

### Official Review · Reviewer_fuyb · 2024-11-04

**Soundness:** 2
**Presentation:** 2
**Contribution:** 2
**Rating:** 5
**Confidence:** 4

**Summary:**

This paper introduces dockedAC, a dataset that leverages molecular docking to generate complex structures associated with each activity value. The dataset incorporates 3D complex structure information, facilitating in-depth study of activity cliffs.

**Strengths:**

The paper is well-written and easy to understand.
The 3D structural information of molecular complexes is crucial for addressing cliff tasks, as varying interaction patterns can significantly impact affinity values. Previous methods for modeling activity cliffs typically focus on feature extraction from the ligand molecule alone, neglecting the surrounding protein environment.
The ablation study of the paper is extensive to reflect the differences between RSME and RMSE_cliff

**Weaknesses:**

For ECFP, what proportion of samples have an ECFP similarity > 0.9, given that similarities > 0.5 are already quite rare?
In pairs of activity cliffs with docking conformations, what noticeable differences exist in their 3D structures? It has not been verified whether docking conformations can accurately capture the true differences in interaction patterns.
Can generated 3D docking conformations, which may differ from the crystal structures, reliably reflect the actual interaction differences between activity cliffs?
In Section 4.3, would training on bioactivity values (pKi/pEC50/pIC50 in log units) together pose any issues?
In Figure 4, although the 3D approach shows a higher RMSE difference and RMSE correlation, the performance gains from the additional 3D complex information are not very significant.
In Table 2, traditional methods (KNN, RF, SVM, etc.), even when only modeling the ligand, still achieve better results, indicating no clear advantage of using 3D complex data.
Overall, I believe that 3D complex structures are valuable for addressing activity cliffs. However, the current 3D complex methods evaluated in this paper, when compared to ligand-only approaches (such as ECFP and 2D graphs), have not shown significant improvement and still fall short of traditional ECFP-based methods. The reasons for this lack of improvement are not quantitatively analyzed, and there is also no evaluation of the quality of docking data to ensure it captures the critical information within activity cliffs.

**Questions:**

See weakness

---

> ### Author Response · Authors · 2024-11-25
> **Thank you for your feedback! (1/2)**
>
> Thank you for your thoughtful review and valuable suggestions. We will address each of your points below.
>
> >For ECFP, what proportion of samples have an ECFP similarity > 0.9, given that similarities > 0.5 are already quite rare?
>
> Thanks for this insightful question. The proportion of pairs with an ECFP similarity > 0.9 is indeed very small, less than 1e-4 out of all the pairs. However, when we annotate the samples appearing in these pairs as AC samples, the proportion of the AC samples represent a significantly larger fraction of all samples, approximately 10%.
>
> Overall, when considering all three similarity metrics, the proportion of the AC samples ranges from 15.7% to 43.2% in our dataset. We will include specific statistics on the proportion of samples with similarity > 0.9. This will help better characterize our dataset's similarity landscape.
>
>
> >In pairs of activity cliffs with docking conformations, what noticeable differences exist in their 3D structures? It has not been verified whether docking conformations can accurately capture the true differences in interaction patterns.
>
> We appreciate your thoughtful comments regarding the reliability of docking conformations in the context of activity cliffs. These are important considerations that merit careful discussion.
>
> To validate our docking protocol, we have compared the docking poses of compounds with known crystal structures in [1]. The observed root-mean-square deviation (RMSD) values (median = 2.55Å) fall well within accepted standards for reliable pose prediction. Building on this foundation, we are implementing template docking strategies for cases where input ligands exhibit high structural similarity to reference ligands, which will further enhance pose accuracy.
>
>
> In the manuscript, we also have discussed the limitations of docking, including potential inaccuracies in conformational sampling and scoring functions. A possible way to make the docking pose better is to run a short molecular dynamics simulation. This would be an important future work for us to enhance the dataset.
>
>
> >Can generated 3D docking conformations, which may differ from the crystal structures, reliably reflect the actual interaction differences between activity cliffs?
>
> We have examined the structural differences in 3D conformations between some activity cliff pairs in [1]. As we have identified the binding site and use a relatively large exhaustiveness parameter for docking, the docking poses for these samples are quite accurate.
>
> We acknowledge your concern about the docking conformations. We have shown that for 3D deep learning models, using the docked strucutral information is able to improve the performance. And as shown in [2], a deep learning model trained on crystal structures can still be used to make predictions with the docked complex. Its performance is quite close to the crystal structure setting because docking error is relatively low.
>
> [1] Husby, Jarmila, et al. "Structure-based predictions of activity cliffs." Journal of chemical information and modeling 55.5 (2015): 1062-1076.
>
> [2] Jin, Wengong, Caroline Uhler, and Nir Hacohen. "SE (3) denoising score matching for unsupervised binding energy prediction and nanobody design." NeurIPS 2024.

---

> > ### Author Response · Authors · 2024-11-25
> > **Thank you for your feedback! (2/2)**
> >
> > >In Section 4.3, would training on bioactivity values (pKi/pEC50/pIC50 in log units) together pose any issues?
> >
> > Thank you for pointing this out. Different bioactivity values (pKi/pEC50/pIC50) are not comparable and cannot be trained together. In our benchmark, we train models for each target separately. We will make it clear in the manuscript.
> >
> > We have added the experiment of training across the targets. We choose to use all the targets with Ki labels. The results can be found in Appendix Figure 13. When trained with multiple targets, the performance is comparable to training with each target separately.
> >
> >
> > >In Figure 4, although the 3D approach shows a higher RMSE difference and RMSE correlation, the performance gains from the additional 3D complex information are not very significant. In Table 2, traditional methods (KNN, RF, SVM, etc.), even when only modeling the ligand, still achieve better results, indicating no clear advantage of using 3D complex data.
> >
> > The good performance of RF, SVM and GBM is mainly due to the handcrafted features they used. For example, ECFPs (Extended-Connectivity Fingerprints) are designed to capture molecular features relevant to molecular activity and are particularly effective for similarity searching and structure-activity relationship (SAR) studies. From Figure 6, it can be seen that the performance of these machine learning models varies with the input features. A drawback of these machine models is that they are target-specific. We discussed potential ways to leverage the strengths of both ECFP-based and 3D-based methods. Our results (as shown in Appendix Table 6 and Figure 8) on ten targets show promising results when combining ECFP with the 3D model IGN.
> >
> > For the deep learning models, the learned features might not be as good as these well-designed handcrafted features. This may be due to the limited amount of available data, which is one of the motivations for our work. We acknowledge that the docking conformations may not be perfect, but our work serves as the first step towards activity cliff research with large-scale 3D complex structures. More data will encourage and enable the deep learning community to develop more effective models for AC prediction.
> >
> >
> > ----
> >
> > We hope that we have properly addressed your concerns. Please let us know if you have any further insights or questions answering which would make you feel more positive about our paper. Thank you again for your time and valuable feedback.

---

> > > ### Author Response · Authors · 2024-11-29
> > >
> > > Dear Reviewer,
> > >
> > > We wanted to follow up on our rebuttal and ensure you had the opportunity to review our responses to your valuable feedback. Your feedback has been instrumental in refining our work. We would greatly appreciate it if you could take a moment to review our rebuttal and consider the clarifications and enhancements we've made. If you find our response satisfactory, we hope you could re-evaluate our paper. We are happy to discuss further if you have other questions.
> > >
> > > Thank you again for your time and consideration.
> > >
> > > Best regards,
> > >
> > > Authors

---

### Meta-Review · Area_Chair_reFa · 2024-12-12

**Metareview:**

The paper introduces a novel benchmark for predicting the bioactivity of molecules focused on the so-called activity cliffs.

Reviewers appreciated the focus on evaluating performance against activity cliffs. A method that can predict accurately how small changes in molecular structure affect activity is likely to have broad applicability and significantly impact on the drug discovery space. In fact, recently deep learning docking methods are under scrutiny due to the reliance on highly similar training data. In general, the benchmark addresses a timely topic.

Reviewers raised some issues regarding the scope of the benchmark. Reviewer sP4z suggested adding recent methods and additional metrics such as ones from the PoseCheck paper.

A significant concern was raised by two reviewers regarding the use of a simple docking method as the ground truth. Indeed, it is not clear if the methods are benchmarked against a high quality or realistic enough target. Relatedly, though this concern wasn't raised by Reviewers, the quality of activity labels in ChEMBL is often low and in AC's opinion the paper doesn't discuss this aspect enough.

Strengths of the paper include also the clarity of writing and the breadth of included models (though AC would find it helpful to include leading pretrained graph transformers such as UniMolv2).

All in all, at the current stage the paper is not yet fully ready for publication. Due to the identified issues by reviewers and AC, there is arguably concern about the impact of the benchmark. These issues are fully solvable, but beyond the scope allowed by the review process of ICLR. I hope Authors will find these comments helpful in strengthening the work.

**Additional Comments On Reviewer Discussion:**

Summarized in the meta-review.

---

### Decision · Program_Chairs · 2025-01-22

Reject